# Dynamics of primitive streak regression controls the fate of neuromesodermal progenitors in the chicken embryo

Charlene Guillot[1,2,3]*, Yannis Djeffal[1,2,3], Arthur Michaut[1,2,3], Brian Rabe[2,4], Olivier Pourquié[1,2,3]*

[1]Department of Pathology, Brigham and Women's Hospital, Boston, United States; [2]Department of Genetics, Harvard Medical School, Boston, United States; [3]Harvard Stem Cell Institute, Boston, United States; [4]Howard Hughes Medical Institute, Boston, United States

**Abstract** In classical descriptions of vertebrate development, the segregation of the three embryonic germ layers completes by the end of gastrulation. Body formation then proceeds in a head to tail fashion by progressive deposition of lineage-committed progenitors during regression of the primitive streak (PS) and tail bud (TB). The identification by retrospective clonal analysis of a population of neuromesodermal progenitors (NMPs) contributing to both musculoskeletal precursors (paraxial mesoderm) and spinal cord during axis formation challenged these notions. However, classical fate mapping studies of the PS region in amniotes have so far failed to provide direct evidence for such bipotential cells at the single-cell level. Here, using lineage tracing and single-cell RNA sequencing in the chicken embryo, we identify a resident cell population of the anterior PS epiblast, which contributes to neural and mesodermal lineages in trunk and tail. These cells initially behave as monopotent progenitors as classically described and only acquire a bipotential fate later, in more posterior regions. We show that NMPs exhibit a conserved transcriptomic signature during axis elongation but lose their epithelial characteristicsin the TB. Posterior to anterior gradients of convergence speed and ingression along the PS lead to asymmetric exhaustion of PS mesodermal precursor territories. Through limited ingression and increased proliferation, NMPs are maintained and amplified as a cell population which constitute the main progenitors in the TB. Together, our studies provide a novel understanding of the PS and TB contribution through the NMPs to the formation of the body of amniote embryos.

*For correspondence:
charlene_guillot@hms.harvard.edu
(CG);
pourquie@genetics.med.harvard.
edu (OP)

Competing interests: The authors declare that no competing interests exist.

## Introduction

The amniote primitive streak (PS), equivalent of the amphibian blastopore, forms at the midline of the embryo during gastrulation. It marks the location where epithelial cells of the epiblast undergo epithelium to mesenchyme conversion and ingress to form the mesoderm and endoderm. At the end of gastrulation, the PS has reached its maximum length and begins to regress, laying in its wake the tissues that will form the embryonic body. When regression is complete, the remnant of the PS morphs into a poorly organized mass of cells located at the posterior end of the embryo, the tail bud, which generates the posterior-most regions of the body (*Stern, 2004*). Fate mapping studies in avian and mouse embryos have shown that at the beginning of PS regression the mesodermal trunk progenitors are found in the epiblast along the PS (*Kinder et al., 1999*; *Psychoyos and Stern, 1996*; *Schoenwolf et al., 1992*; *Smith et al., 1994*; *Spratt and Condon, 1947*; *Tam and Beddington, 1987*; *Wilson and Beddington, 1996*). These studies demonstrated that the anteroposterior distribution of mesodermal progenitors in the PS epiblast reflects their future medio-lateral fate. Cells within the node generate the notochord, and the most anterior PS cells produce the paraxial

mesoderm, while the territories of the intermediate, lateral plate and extraembryonic mesoderm lie in progressively more posterior regions of the PS. The recent discovery of a new population of axis progenitor stem cells, the neuromesodermal progenitors (NMPs) in mouse, has led to reconsider the PS role in amniote body formation (*Tzouanacou et al., 2009*). These bipotent stem cells generating large clones containing both neural and mesodermal derivatives along the forming body axis were identified by a retrospective strategy, which did not allow to precisely localize them in the embryo. Intriguingly, direct reconstruction of the epiblast cell lineage from high-resolution light-sheet imaging of developing mouse embryos did not identify bipotential NMP cells in the PS region but only cells fated to one or the other lineage (*McDole et al., 2018*). Similar retrospective tracking experiments based on time-lapse movies of fluorescent cells in transgenic chicken embryos expressing GFP have led to identify a small population of epiblast cells that gives rise to descendants lying in neural and mesodermal territories (*Wood et al., 2019*). In the chicken and mouse embryo, extensive fate mappings of the PS region have been performed, but they did not reveal the existence of NMPs giving rise to paraxial mesoderm and neural tube (*Brown and Storey, 2000*; *Iimura et al., 2007*; *Psychoyos and Stern, 1996*; *Schoenwolf et al., 1992*; *Selleck and Stern, 1991*; *Tam and Beddington, 1987*; *Wilson and Beddington, 1996*; *Wilson et al., 2009*). Grafts of small territories of the epiblast adjacent to the anterior PS in avian or mouse embryos can give rise to both neural and mesodermal derivatives, suggesting that this territory could contain NMPs (*Garcia-Martinez et al., 1993*; *Iimura et al., 2007*; *Wymeersch et al., 2016*). In mouse, cells of this region (caudal lateral epiblast and node streak border) coexpress the neural marker Sox2 and the mesodermal marker T/Brachyury and were proposed to include the NMP population (*Wymeersch et al., 2016*). However, these grafting experiments do not allow to distinguish between a population of bipotential cells and a mixture of precursors committed to each lineage. To our knowledge, the only direct evidence for epiblast precursors able to give rise to both neural and mesodermal lineages in amniotes comes from single-cell injections of horseradish peroxidase in the mouse epiblast followed by embryo culture (*Forlani et al., 2003*). In this work, six clones with descendants in both lineages have been obtained from injections between the late streak and head fold stage. However, this study was performed before identification of NMPs by retrospective cloning and the bifated clones are not discussed in the article. Thus, the localization and fate of NMPs in amniotes remain incompletely understood.

Here, we performed direct lineage analysis of cells in the anterior PS epiblast of chicken embryos using two different approaches: labeling cells with a barcoded retroviral library and a Brainbow-derived strategy (*Loulier et al., 2014*). We show that single cells of the SOX2/T-positive region of the anterior PS are NMPs that can contribute to both neural tube and paraxial mesoderm and self-renew during formation of more posterior regions of the body. As most published fate mappings have been analyzed only after short term and did not study formation of the posterior regions of the embryo, cells of the anterior epiblast were considered monopotent and their bipotentiality went undetected. Thus, our findings reconcile the existence of bipotential NMPs demonstrated in mouse with the large body of amniote fate mappings described in classical developmental biology literature. We also performed single-cell RNA-sequencing (scRNAseq) analyses of the region encompassing the anterior PS epiblast and tail bud during axis elongation. This led to the identification of a cluster exhibiting two developmental trajectories leading to neural and mesodermal fates as expected for NMPs. We identified a similar cluster from published scRNAseq datasets from posterior tissues of equivalent stages of developing mouse embryos. In mouse and chicken embryos, these NMP populations maintained a conserved transcriptional identity segregating from their neural and mesodermal descendants as a single-cell cluster. During formation of the posterior body, the NMP population nevertheless underwent maturation characterized by expression of posterior Hox genes and loss of the epithelial phenotype. Moreover, using in toto imaging, we show that a posterior-to-anterior gradient of cell convergence speed toward the PS coupled to ingression results in the progressive exhaustion of the PS from its posterior end. This leads to the successive disappearance of the territories of the extraembryonic, lateral plate, intermediate mesoderm, and non-NMP paraxial mesoderm precursors (PMPs). Increased proliferation combined with limited ingression of cells ensures the self-renewal and expansion of the territory of clonally related bipotential NMPs during body formation, which constitutes the last remnant of the PS to contribute to the tail bud. Thus, our work provides a direct demonstration of the existence of NMP cells in amniotes and a mechanistic understanding for their sustained contribution to axis elongation.

## Results

### Characterization of the SOX2/T-positive territory of the epiblast and the anterior PS region

As co-expression of SOX2 and T has been proposed to identify NMPs in mouse embryos (*Wymeersch et al., 2016*), we first set out to characterize the double-positive SOX2/T population in the epiblast during development of the chicken embryo. At stage 4⁻ HH (*Hamburger and Hamilton, 1992*), SOX2 is only expressed in the neural plate in the anterior epiblast and its expression domain progressively expands posteriorly to reach the anterior PS and the adjacent epiblast at stage 4-5HH (*Figure 1A*, *Figure 1—figure supplement 1A–C*). SOX2 expression levels show a graded pattern, which peaks in the anterior neural plate and decreases in the epiblast lateral to the anterior-most PS (*Figure 1A*, *Figure 1—figure supplement 1D–F*). No SOX2 expression is detected in the epiblast adjacent to the posterior PS. At these stages, T expression shows a reverse gradient along the PS, with low levels in the epiblast around the node region and high levels in the posterior end of the PS (*Figure 1B*, *Figure 1—figure supplement 1E, F*). At stage 4+HH, these opposing gradients begin to overlap in the epiblast lateral to the anterior PS, defining a SOX2/T territory that forms an inverted V capping the PS (*Figure 1C, D*, *Figure 1—figure supplement 1G, H*). At stage 5HH, numerous double-positive SOX2/T cells are found in the epiblast of the anterior PS region, with the underlying mesoderm expressing the paraxial mesoderm-specific marker MSGN1 (*Figure 1E, F, E', F'*, levels 2 and 3, *Figure 1—figure supplement 1G, H*). Posterior to this domain, the epiblast of the PS region expresses T but not SOX2, while the underlying mesoderm maintains a high level of MSGN1 expression (*Figure 1E, F, E'', F''*, level 4). Posterior to these two domains, in regions of the epiblast corresponding to the prospective intermediate mesoderm/lateral plate and the extraembryonic mesoderm (*Psychoyos and Stern, 1996*), epiblast cells express T but not SOX2 and no MSGN1-positive cells are found in the underlying mesoderm (*Figure 1E, F*, level 5). Therefore, the presumptive paraxial mesoderm territory of the epiblast of the anterior PS can be subdivided into an anterior domain where cells co-express SOX2 and T corresponding to the presumptive NMP territory and a posterior PMP domain where cells express T but not SOX2 (*Figure 1D, E', E''*). Both domains are characterized by the production of MSGN1-positive PMPs.

### Cells of the anterior primitive streak epiblast contribute to the neural tube and paraxial mesoderm tissues during axis formation

In order to explore the fate of cells of the SOX2/T-positive region of the anterior PS, we first used a library of barcoded defective retroviruses expressing GFP (*Harwell et al., 2015*) to infect the epiblast of the anterior PS region at stage 5HH (*Figure 2A, B*). Embryos were reincubated for 36 hr, and single fluorescent cells were manually harvested from the paraxial mesoderm and neural tube from embryo sections for subsequent barcode analysis. We identified seven clones expressing unique barcodes (*Figure 2C*). Four clones contained cells both in the neural tube and in somites/presomitic mesoderm (PSM), indicating the bipotential nature of the infected cells of the epiblast of the anterior PS region. Descendants of bipotent cells were found in both anterior (before somite 27, which marks the transition between primary and secondary neurulation; *Le Douarin et al., 1998*), and posterior regions of the axis (*Figure 2B*). To confirm these observations, we performed lineage tracing of the SOX2/T region of the epiblast using genetic labeling based on the Brainbow-derived MAGIC markers (*Loulier et al., 2014*). To mark cells in the SOX2/T domain, we co-electroporated plasmids expressing a self-excising Cre recombinase and the Nucbow transgene together with the TolII transposase to drive transgene integration. Electroporation of this set of constructs allows to permanently mark cell nuclei with a specific color code generated by the unique combination of different fluorescent proteins triggered by random recombination of the Nucbow cassette (*Loulier et al., 2014*). This color code is then stably transmitted to each daughter cell and can be retrieved by confocal imaging and quantification of the color hues. Using very fine electrodes, we could electroporate as low as 10 epiblast cells of the anterior PS region at stage 5HH (*Figure 2—figure supplement 1*). We harvested embryos 36 hr (stage 17HH, *Figure 2B, D*) and 56 hr (stage 20HH, *Figure 2H, I*) after electroporation. We identified 47 clones containing a total of 690 cells in the neural tube and paraxial mesoderm in six embryos (*Figure 2D–K*). While we found both monopotent neural and mesodermal clones, the majority of the clones were bipotent (*Figure 2E–G, J, K*).

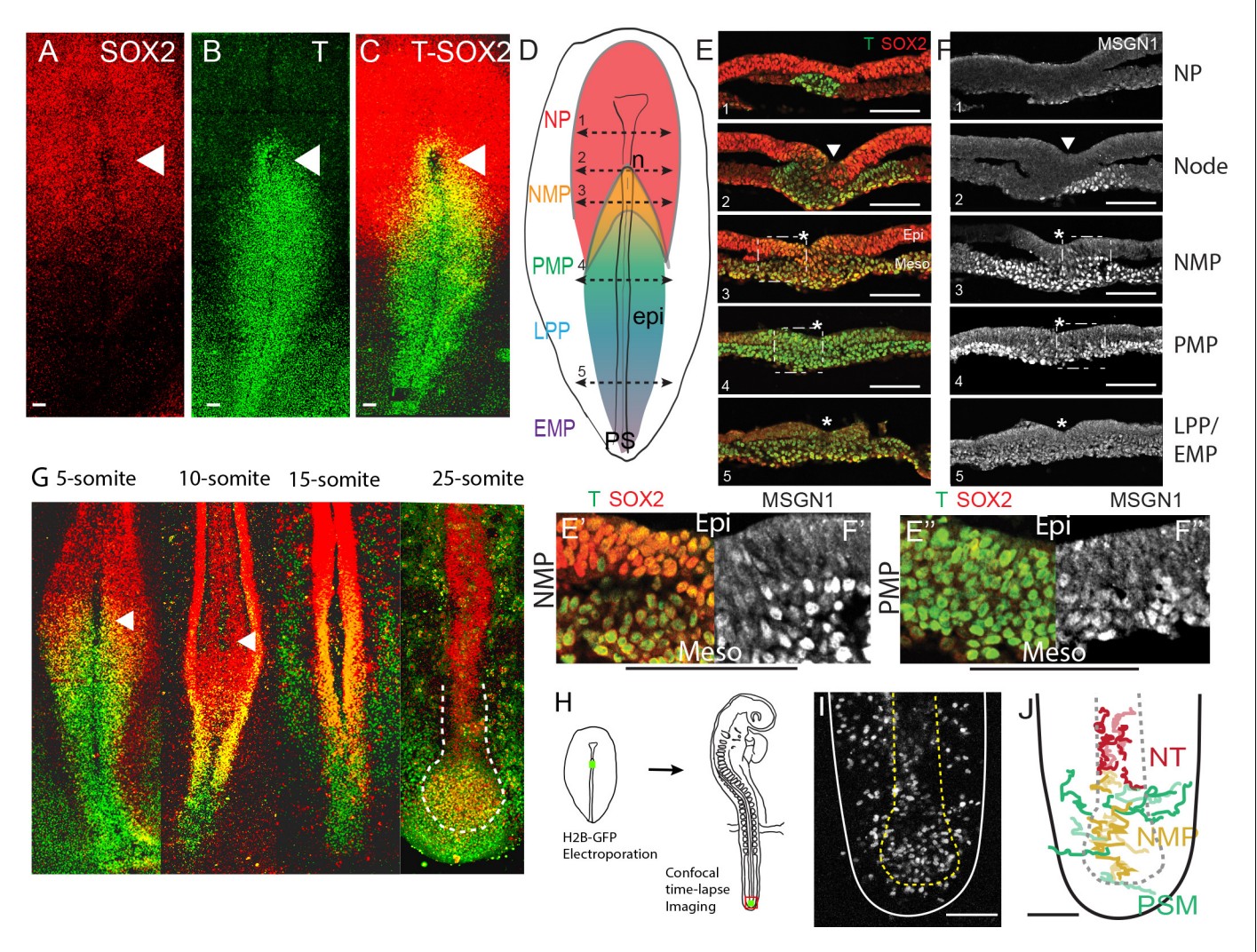

**Figure 1.** Characterization of the SOX2/T-positive territory of the epiblast. (A–C) Whole-mount embryos and (E, F) transverse cryosections showing the immunolocalization of SOX2 (A), T (B), T and SOX2 (C, E), and MSGN1 (F) in chicken embryos at stage 5HH. (D) Schematic representation of the expression of T (blue: high; green: low), SOX2 (red), and SOX2/T (gold) in a stage 5HH chicken embryo. The level of the tissue sections in (E, F) is shown with dashed double arrows labeled from 1 to 5 from anterior to posterior. (E', F') Higher magnification of the NMP region (level 3, E, F). (E'', F'') Higher magnification of the PMP region (level 4, E, F). (n = 7 embryos for whole mount; n = 3 for cryosections). PS regions are defined based on distance from Hensen's node as described in *Psychoyos and Stern, 1996*. (G) Maximum intensity projections from confocal images of chicken embryos immunostained for T (green) and SOX2 (red) proteins. Double-positive cells are shown in yellow. White hatched line in the 25-somite embryo marks the end of the neural tube (red) and the NMP region (orange) (n = 23 embryos analyzed in total). (H) Diagram summarizing the experimental procedure to label NMP cells in stage 5HH embryos using electroporation of a fluorescent reporter in the epiblast of the anterior PS region (in green) followed by analysis at the tail bud stage. (I, J) Fate of descendants of cells of the NMP region electroporated at stage 5HH with an H2B-RFP plasmid and imaged in time lapse at the 25-somite stage in the tail bud region. Z-projection from confocal images (I) and tracks (J) of a time-lapse movie showing the movements of the cells in the NMP territory for 10 hr (*Video 1*). Tracks were color-coded a posteriori. Neural, mesodermal, and NMP cell trajectories are shown in red, green, and gold, respectively. NP: neural plate; NMP: neuromesodermal progenitors; PMP: presomitic mesoderm progenitors; LPP: lateral plate progenitors; EMP: extraembryonic mesoderm progenitors; n: node; epi: epiblast; Meso: mesoderm; PS: primitive streak; NT: neural tube. Arrowheads: Hensen's node. Asterisk: primitive streak. (A–D, G–J) Dorsal views. Anterior to the top. Scale bar: 100 μm.

The online version of this article includes the following source data and figure supplement(s) for figure 1:

**Figure supplement 1.** Onset of SOX2/T expression cells in chicken.

**Figure supplement 1—source data 1.** Number of T/SOX2 double-positive cells during early chicken stages.

**Figure supplement 2.** Fate of cells of the anterior PS region at the tail bud stage.

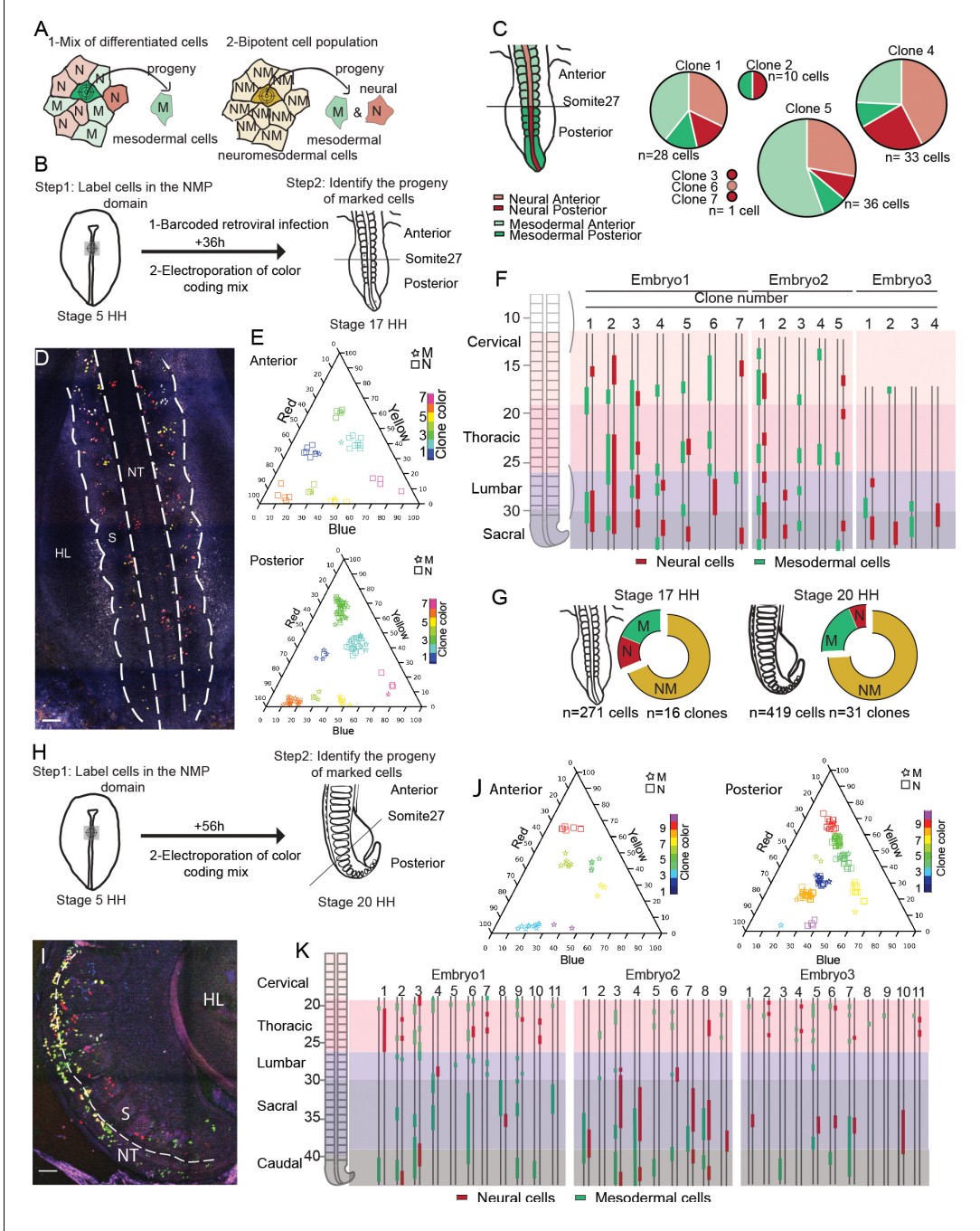

**Figure 2.** Cells of the SOX2/T-positive territory of the anterior primitive streak epiblast contribute to the neural tube and paraxial mesoderm tissues during axis formation. (**A**) Schematic diagram showing the strategy used to decipher if the neuromesodermal progenitor (NMP) territory is a mix of monopotent cells (left) or composed of bipotent cells (right). Schemes show an example of a cell that has been marked by retroviral barcoding or genetic color coding and its expected outcome in the different cases (arrows). The color indicates the neural (red, N), the mesodermal (green, M), and the neuromesodermal (gold, NM) identities. (**B**) Experimental procedure showing the infected or electroporated region of the epiblast at stage 5HH (left, green) and the stage at which embryos were harvested for analysis (n = 3). (**C**) (left) Diagram showing the neural tube (red) and paraxial mesoderm (green) in the anterior (light) and posterior (dark) regions of the embryo. (Right) Pie graphs showing the distribution of the neural (red) and mesodermal (green) cells anterior (light) or posterior (dark) to the 27th somite in the seven clones identified by retrovirus labeling analyzed (n = 110 cells in three embryos). (**D**) Confocal z-section showing the region of a stage 17HH embryo shown in (**H**) and acquired using three separated laser paths to retrieve the color codes genetically encoded as described in *Loulier et al., 2014*. (**E**) Triplot diagrams showing the distribution of descendants of cells labeled with different Nucbow combinations in the anterior (top) and posterior (bottom) regions of seven clones in a representative stage 17HH embryo. Each symbol represents a cell identified based on the percentage of red, blue, and yellow expressed. The symbols are colored based on their clonal identity.

*Figure 2 continued on next page*

*Figure 2 continued*

Squares: neural cells; stars: mesodermal cells. (F) (left) Region analyzed showing the different axial levels. (Right) Axial distribution of the clones in three-stage 17HH embryos. Red bars: neural cells; green bars: mesodermal cells. (G) Quantification of the different clones: mesodermal (M, green), neural (N, red), and bipotent neuromesodermal clones (NM, gold) at stage 17HH (left) and stage 20HH (right) (n = 16 clones, 271 cells in three embryos) and (n = 40 clones, 519 cells in three embryos), respectively. (H) Experimental procedure showing the electroporated region of the epiblast at stage 5HH (left, green) and the stage at which embryos were harvested for analysis (n = 3). (I) Confocal z-section using three-color imaging (*Loulier et al., 2014*) corresponding to the posterior region of a stage 20HH embryo shown in (H). (J) Triplots showing the distribution of 10 representative clones in the anterior (left) and posterior (right) regions of a stage 20HH embryo electroporated at stage 5HH. Squares: neural cells; stars: mesodermal cells. (K) (left) Region analyzed showing the different axial levels. (Right) Axial distribution of the clones in three embryos. Green bars: mesodermal cells; red bars: neural cells, double line: anteroposterior axis. M: mesoderm; N: neural; NM: neuromesodermal; S: somite; HL: hindlimb; D: dorsal views. Anterior to the top. Scale bar: 100 µm.

The online version of this article includes the following source data and figure supplement(s) for figure 2:

**Source data 1.** Retrovirus and Brainbow labeling of chicken embryo.
**Source data 2.** Matlab code for clone identification.
**Figure supplement 1.** Quantification of the number of epiblast cells electroporated.
**Figure supplement 1—source data 1.** Number of electroporated cells after one or two pulses.

Bipotent clones were found both in the anterior and posterior region, and they often exhibit descendants of only one lineage in some regions and of the other lineage in other regions (*Figure 2F, K*). Thus, our lineage-tracing analysis provides direct evidence for the existence of bipotent NMP cells located in the SOX2/T region of the anterior PS epiblast in the chicken embryo.

We next investigated the fate of the SOX2/T territory during axis formation. During PS regression, that is, up to the 10–12-somite stage, the SOX2/T cells were maintained in the epiblast lateral to the anterior-most part of the PS, below Hensen's node (arrowhead, *Figure 1G*). After the 10-somite stage, cells were found in continuity with the posterior-most SOX2-positive/T-negative neural tube (*Figure 1G*). These SOX2/T cells eventually became located in a superficial region of the tail bud at the 25-somite stage where they remain at least until stage 26HH (*Olivera-Martinez et al., 2012*; *Figure 1G*). In order to analyze the lineage continuity of cells of the anterior PS region, we performed local electroporations of small groups of epiblast cells with a H2B-RFP reporter in ovo, targeting the SOX2/T-positive territory of the anterior PS region at stage 5HH (*Figure 1H*). We next performed confocal live imaging of the labeled embryos to track the RFP-expressing cells at the 25-somite stage (*Video 1*). At this stage, electroporated cells were found in the neural tube, paraxial mesoderm, and superficial region of the tail bud where the SOX2/T cells were identified (*Figure 1G, I*, *Figure 1—figure supplement 2A, B*). Thus, this tail bud territory contains descendants of the SOX2/T cells of the anterior PS epiblast region of stage 5HH embryos. We next performed time-lapse imaging to track fluorescent cells from this superficial SOX2/T-positive territory to examine

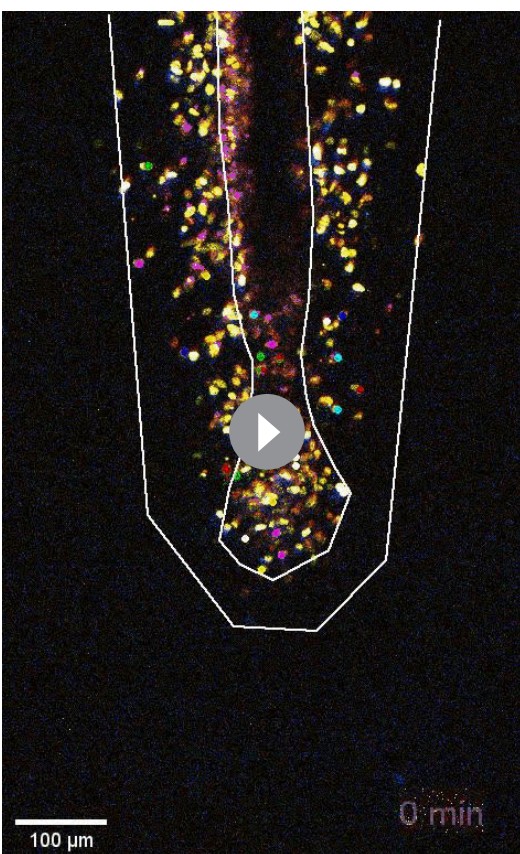

100 µm 0 min

**Video 1.** Tracking descendants of the anterior primitive streak (PS) epiblast at 25 somites. Time-lapse movie of dorsal maximum z-projections showing descendants from the neuromesodermal progenitor (NMP) region electroporated at stage 5HH with an H2B-RFP plasmid in time lapse. t = 4 min between frames, movie = 10 hr. z-sectioning = 4 µm. 20× objective, LSM 780.
https://elifesciences.org/articles/64819#video1

their fate (*Figure 1H–J*, *Videos 1* and *2*). We observed anterior cells undergoing limited movements along the AP axis and moving to join the neural tube (*Figure 1J*, red tracks, *Videos 1* and *2*). These cells subsequently acquired the characteristic mediolateral elongated shape of neural tube cells (*Figure 1—figure supplement 2C*, *Video 2*). Cells in the middle of the SOX2/T territory moved in the AP direction but undergo very limited medial to lateral movements (*Figure 1J*, gold tracks, *Figure 1—figure supplement 2D*, *Videos 1* and *2*). These cells neither joined the neural tube nor the mesoderm, suggesting that they remained in a progenitor state. In contrast, some posterior cells undergo dorsal to ventrolateral movements into the mesoderm (*Figure 1J*, green tracks, *Figure 1—figure supplement 2E*, *Videos 1* and *2*). Therefore, our data supports lineage continuity and conserved bipotential fate of the SOX2/T territory during axis elongation, suggesting that NMPs constitute a stem cell population able to contribute to the neural and mesodermal fates and to self-renew.

## scRNAseq analysis of the precursors of posterior tissues during axis formation in chicken and mouse

To characterize the molecular identity of these NMPs, we performed scRNAseq of cells dissociated from a micro-dissected region encompassing the anterior PS in stage 5HH and in 6-somite embryos as well as the tail bud of 35-somite embryos (*Figure 3A–C*). We used the inDrops sequencing platform to analyze 2059 cells at stage 5HH, 1628 cells at the 6-somite, and 3561 cells at the 35-somite stage. We identified clusters that could be assigned expected cell identities of the dissected regions for each developmental stage based on the expression of specific genes (*Figure 3A–C* and *Supplementary files 1* and *2*, *Figure 3—figure supplement 1*). Expression of the genes at the corresponding stages was validated based on the Geisha In Situ hybridization database (http://geisha.arizona./). Heat maps showing the top differentially expressed genes used to identify the clusters are shown in *Figure 3—figure supplement 1*, and the lists of differentially expressed genes are shown in *Supplementary file 2*.

At stage 5HH, we identified a large cluster corresponding to epiblast cells, characterized by the expression of *SALL4* or *FRZB*. Another cluster contained cells co-expressing markers of paraxial mesoderm (*MSGN1*, *MEOX1*) and lateral plate (*TWIST1*, *GATA5*), suggesting that they correspond to ingressed mesoderm contributing to the anterior-most somites as recently shown in mouse embryos (*Guibentif et al., 2021*). A cluster containing early neural plate cells characterized by expression of *SFRP2* and *PDGFA*, as well as small clusters corresponding to the endoderm (*SOX17*, *FOXA2*) and the notochord (*CHRD*, *NOTO*), was also identified. The last cluster (yellow) expressing genes such as *TBXT*, *CDH2*, *GJA1*, and *WNT8A* is found in between neural and mesodermal clusters, suggesting that it could represent the NMP population (*Figure 3A*). At the 6-somite stage, the large epiblast cluster was not present anymore. We identified a cluster of cells expressing PSM markers such as *MSGN1* or *TCF15* but no lateral plate markers, suggesting that they are precursors of more posterior somites. There were also two clusters of cells expressing genes associated with posterior PS (*MSX2*, *JAM3*, *BAMBI*) and LP (*PITX2*, *GATA2*) identities. We also identified clusters with neural (*HES5*, *PDGFA*), notochord (*NOTO*), and endoderm (*SOX17*) identities. A large cluster of cells connected to the neural and PSM clusters represents putative NMPs (*Figure 3B*). At the 35-somite stage, cell populations of expected cell fates were clearly segregated, with clusters of LP (*MSX2*, *PRRX1*), blood (*HBZ*), notochord (*CHRD*), neural (*HES5*, *PAX6*), and ectoderm (*EPCAM*, *WNT6*). Cells with a PSM (*MSGN1*, *DLL1*) and somitic identity (*MEOX1*, *TCF15*) also formed separate clusters. As for the earlier stages, there was a putative NMP cluster

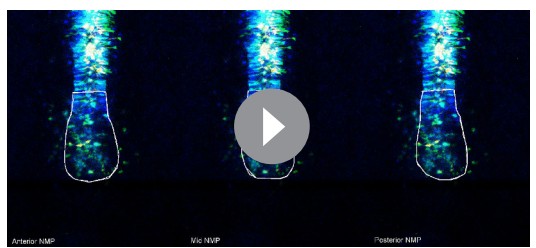

**Video 2.** Tracking the fate of descendants of the anterior epiblast at 25 somites. Time-lapse movie of dorsal maximum z-projections of a 25-somite embryo showing the localization and fate of descendants from cells of the anterior primitive streak (PS) epiblast region co-electroporated at stage 5HH with a GAP43-Venus and an H2B-RFP plasmid (marking the membrane and the nucleus, respectively). The SOX2/T region of the tail bud is delimited by the white lines. Tracks of selected cells are shown. Color code indicates the z position of the cells (yellow: dorsal; blue: ventral). t = 8 min between frames. z-sectioning = 4 μm. 20× objective, LSM 780.
https://elifesciences.org/articles/64819#video2

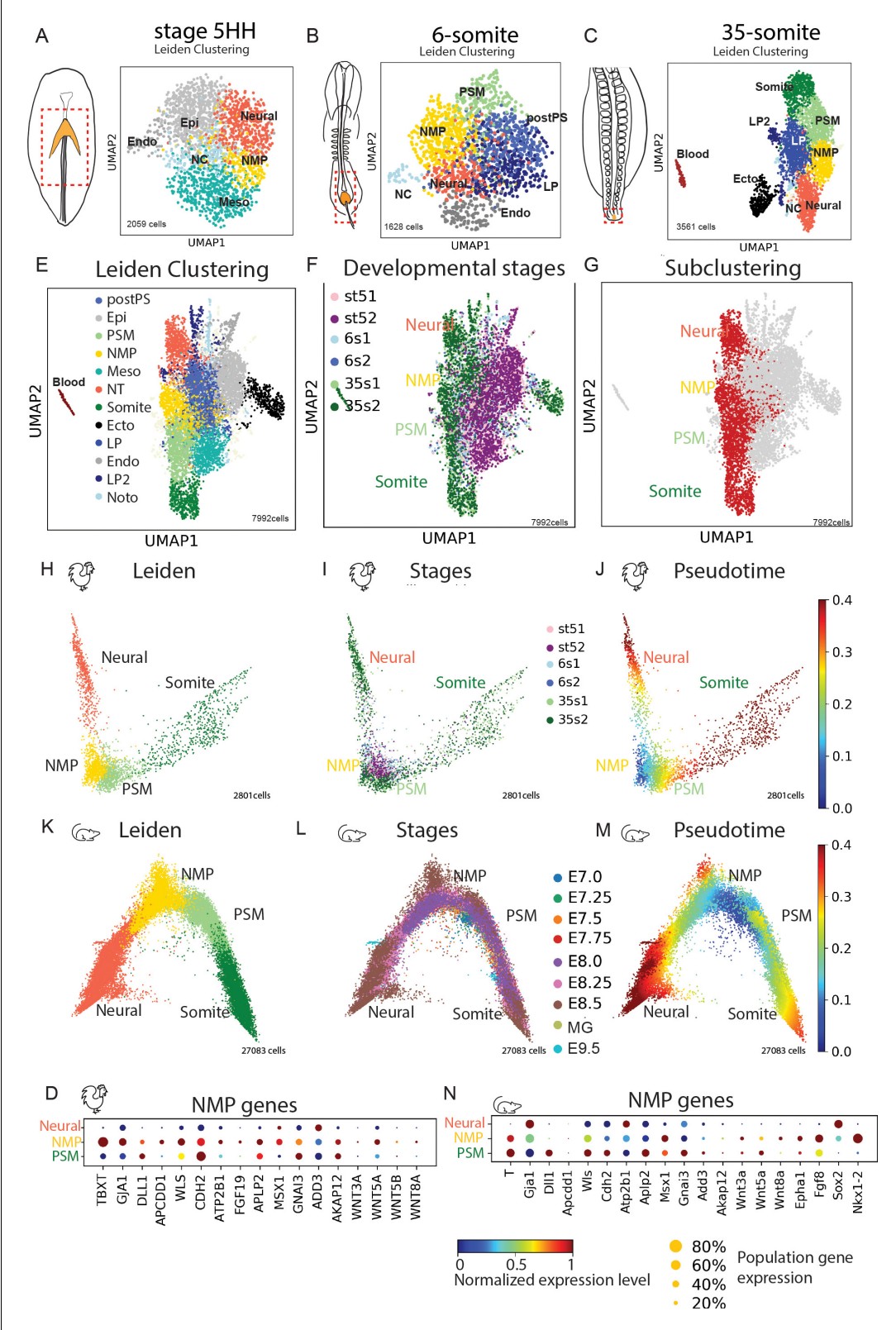

**Figure 3.** Single-cell RNA-sequencing (scRNAseq) analysis of the posterior tissue precursors during posterior axis formation. (**A–C**) (left) Diagrams of a stage 5HH (**A**), a 6-somite (**B**), and a 35-somite (**C**) chicken embryo showing the region dissected and analyzed by scRNAseq in the hatched red boxes, which includes the NMP territory (in gold). (Right) *k*-NN graph showing the 2059 cells sequenced from stage 5HH embryos (**A**), the 1628 cells sequenced from 6-somite embryos (**B**), and the 3561 cells sequenced from 35-somite embryos (**C**) visualized with Uniform Manifold Approximation and

*Figure 3 continued on next page*

Figure 3 continued

Projection (UMAP). Single cells are colored based on Leiden clustering identities. (D) Dotplot showing the expression level of NMP signature genes in the chicken neural, NMP, and PSM clusters. (E–G) k-NN graphs combining all sequenced cells from stage 5HH, 6-somite and 35-somite chicken embryos (total: 7992 cells), visualized with UMAP and colored following Leiden clustering to show cell identities (E), or by developmental stage (F) or to show cells of the neural, NMP, PSM, and somite clusters (in red), which were used for the subsequent analysis shown in (H–J). Note that cells in the tan color belong to different cluster identities with less than 50 cells, and we decided to show them to represent the entirety of the data but do not analyze them in our study. (H–J) k-NN graphs showing cells of the chicken neural, NMP, PSM, and somite clusters extracted based on the analysis shown in (E–G) (total: 2801 cells) from stage 5HH, 6-somite, and 35-somite visualized with diffusion map (diffmap) and analyzed using Leiden clustering to show cell identities that include neural, NMP, PSM, and somite (H), as well as the developmental stage (I) or pseudo-temporal ordering with the NMP cluster as the starting node (J). (K–M) k-NN graphs showing cells of the mouse neural, NMP, and PSM clusters extracted based on the analysis shown in *Figure 3—figure supplement 3* (total: 27,083 cells) from stage E7.0–E9.5 visualized with diffusion map (diffmap) and analyzed using Leiden clustering to show cell identities that include neural, NMP, and PSM (K), as well as the developmental stage (L) or pseudo-temporal ordering with the NMP cluster as the starting node (M). (N) Dotplot showing the expression level of NMP signature genes in mouse in the neural, NMP, and PSM clusters. Ecto: ectoderm; Endo: endoderm; Epi: epiblast; Meso: mesoderm; LP: lateral plate; NMP: neuromesodermal progenitors; NT: neural tube; NC: notochord; PSM: presomitic mesoderm; postPS: posterior PS; SOM: somite; LP2: lateral plate 2; MG: mixed gastrulation in (D, N). Circle size shows the percentage of cells expressing the gene in the cluster. Color shows the normalized level of expression. Normalization is done by gene across the clusters.

The online version of this article includes the following figure supplement(s) for figure 3:

**Figure supplement 1.** Single-cell RNA-sequencing (scRNAseq) analysis of stage 5HH to 35-somite chicken embryos.

**Figure supplement 2.** Analysis of combined data from stage 5HH, 6-somite, and 35-somite chicken embryos and identification of chicken neuromesodermal progenitor (NMP) signature genes.

**Figure supplement 3.** Analysis of combined data from E7.0 to E9.5 mouse embryos to identify the neuromesodermal progenitor (NMP), presomitic mesoderm (PSM), neural, and somite clusters.

**Figure supplement 4.** Expression of neuromesodermal progenitor (NMP) signature genes in chicken and mouse.

lying between the neural and PSM clusters (*Figure 3C*). Thus, at the three developmental stages, we observed a potential NMP cluster showing an identity different from ingressed mesoderm, neural plate, endoderm, and epiblast (*Figure 3A–C*). We identified a signature of genes that are differentially expressed in this cluster and conserved in the three potential NMP clusters. These include *TBXT, GJA1, DLL1, APCDD1, WLS, CDH2, ATP2B1, FGF19, APLP2, WNT5A, GAD1, PALD1, AKAP12, EPHA1,* and *WNT8A* (*Supplementary file 2*, *Figure 3—figure supplement 2*). The majority of these genes are either targets or members of the Wnt pathway, which is critical for NMP differentiation in vivo and in vitro (*Henrique et al., 2015*).

We next used the Leiden algorithm to perform a clustering analysis of the three developmental stages together (*Figure 3E, F*, *Figure 3—figure supplement 3*). This also led to the identification of clusters matching the well-known cell identities of the posterior region described above (*Supplementary file 2*), including a putative NMP cluster between the PSM and the NT clusters (*Figure 3E*, *Figure 3—figure supplement 2B*). We extracted and reanalyzed 2801 cells of the NMP, neural tube, and paraxial mesoderm clusters (in red, *Figure 3G*). Using Leiden clustering algorithm, we found a unique NMP cluster composed of cells from all three developmental stages, lying in between clusters of cells of the NT and the PSM (*Figure 3H, I*). Using a linear discriminant analysis (LDA) classifier, we confirmed that identified clusters are related to clusters of similar identity from the posterior region of an E9.5 mouse embryo (*Diaz-Cuadros et al., 2020*; *Figure 3—figure supplement 2C*). An analysis of the pseudo-temporal developmental trajectory of NMPs shows two major differentiation paths leading either to neural or mesodermal fate, thus supporting the bipotentiality of these cells (*Figure 3J*, *Figure 3—figure supplement 2D*). Altogether, our data suggest that chicken NMPs are maintained as a single-cell population with a distinct transcriptional identity during axis elongation.

We next performed a parallel analysis in mouse, combining a mouse embryo scRNAseq dataset of embryos ranging from E7 to E8.5 (116,312 cells) (*Pijuan-Sala et al., 2019*) with our data of an E9.5 mouse embryo posterior region (4367 cells) (*Diaz-Cuadros et al., 2020*) in order to cover a similar developmental period to our chicken scRNAseq data. Clustering analysis combined with examination of the top differentially expressed genes in the clusters identified NT and paraxial mesoderm clusters (PSM and somite) as well as a distinct cluster lying in between, in which cells express *T, Sox2,* and *Nkx1.2*, suggesting that these cells are NMPs (*Gouti et al., 2017*; *Figure 3—figure supplement 3A–C*, *Supplementary file 2*). We next extracted cells of the NMP, neural, and PSM/somite clusters from this dataset to analyze them separately. Leiden-based reclustering analysis led to the

identification of NT, NMP, PSM, and somite clusters (*Figure 3K*). The NMP cluster includes cells from all the developmental stages analyzed starting from E7.0 up to E9.5, indicating that these cells are maintained as a stable population during axis elongation (*Figure 3L*, *Supplementary file 1*). Using pseudotime analysis, we identified two main developmental trajectories arising from the NMP cluster and leading to neural and mesodermal identities, supporting the bipotentiality of these cells (*Figure 3M*, *Figure 3—figure supplement 3E*). This is consistent with previous reports indicating that NMPs first form at E7.0 in mouse embryos (*Gouti et al., 2017*; *Guibentif et al., 2021*), that is, at a roughly equivalent stage to chicken embryos (*Figure 1—figure supplement 1*), which corresponds to the beginning of PS regression. Most NMP signature genes common to the three chicken NMP clusters were also expressed in the mouse NMP cluster (*Figure 3D, N*, *Figure 3—figure supplement 4*). A third developmental trajectory was also observed within the NMP cluster, correlating with the age of the embryo (*Figure 3K–M*). This suggests that while they exhibit a largely conserved transcriptional identity, cells of the NMP cluster nevertheless show some maturation during axis elongation. This trajectory is also visible in the chicken pseudotime analysis although it is less conspicuous (*Figure 3J*).

Altogether, our data identify NMPs as a cell population conserved in chicken and mouse that exhibit a distinct identity from mesodermal and neural cells. These cells first become specified at the beginning of PS regression and remain as a population of bipotential precursors contributing both to the neural and mesodermal territories during formation of the posterior embryonic axis.

## Quantitative analysis of the transition states between NMP, neural, and PSM fates in the mouse and chicken embryo

We next used a trajectory inference technique based on optimal-transport (Waddington-OT [WOT]) to analyze quantitatively how NMP cells transition between the NMP, PSM, and neural states (*Schiebinger et al., 2019*). This method allows to compute the probability distribution in gene-expression space where each cell has a distribution of both probable origins and probable fates. By inferring the temporal couplings with optimal transport of the cells between NMPs, PSM, and the neural tube, we reconstructed their probabilistic developmental trajectories in a transport map. The trajectories represent the probability vector for a cell to join the cluster of interest at day E9.5 in the mouse dataset and at 35 somites in the chicken dataset. We first extracted cells from mouse and chicken NMP clusters as well as their expected descendants of the PSM and neural tube from the clusters identified above. Following batch correction, Principal Component Analysis (PCA), and Uniform Manifold Approximation and Projection (UMAP) projection, cells were clustered using Leiden, leading to the identification of NMP, PSM, and neural tube clusters (*Figure 4A, B*). We next applied WOT to this restricted dataset. This approach identified PSM and neural clusters as NMP descendants, while cells of the NMP cluster were classified as ancestors to both the neural and PSM cellular states (*Figure 4A, B*). This strategy identified two trajectories from NMP to PSM and from NMP to NT reflecting the expected fate of these cells during development. It also showed a third trajectory within the NMP cluster, supporting the self-renewal and maturation of these cells during development (*Figure 4A, B*). These results identify bifated neural and mesodermal precursors within the same gene-expression space. Now that we identified the cell sets at both the beginning (NMP) and ending timepoints (neural and mesodermal), we computed the transition table showing the transported mass from the NMP cells toward the NMP, neural, and PSM cellular states (*Figure 4C, D*). We find that NMPs can self-maintain by contributing to the NMP cell state itself (36% in chick and 37% in mouse), suggesting that they are able to self-renew. The NMP cells also transition toward the neural (33% in chick and 36% in mouse) and mesodermal states (31% PSM in chick and 27% in PSM).

We also studied the transcription factor gene sets underlying the neural and mesodermal trajectories (*Figure 4E*). The WOT analysis identified transcription factors that are enriched in cells most likely to transition to each particular fate (*Supplementary file 3*). The limited sequencing depth of the chicken dataset severely limited the resolution of the analysis. Thus we focused on the mouse dataset in which we identified transcription factors that can predict the different fates analyzed. We confirmed that the expression trends of these genes are showing a coherent progression both in mouse and chicken datasets and generated a curated list of the putative transcription factors associated to NMP maturation (gold), PSM (green), and neural tube (red) trajectories (*Figure 4E–G*, *Supplementary file 3*). We identified transcription factors that are both up- and downregulated in the NMP trajectory in both species. These include Rarg, Tbxt, Etv5, Mnx1, Msx1, Evx1, which were

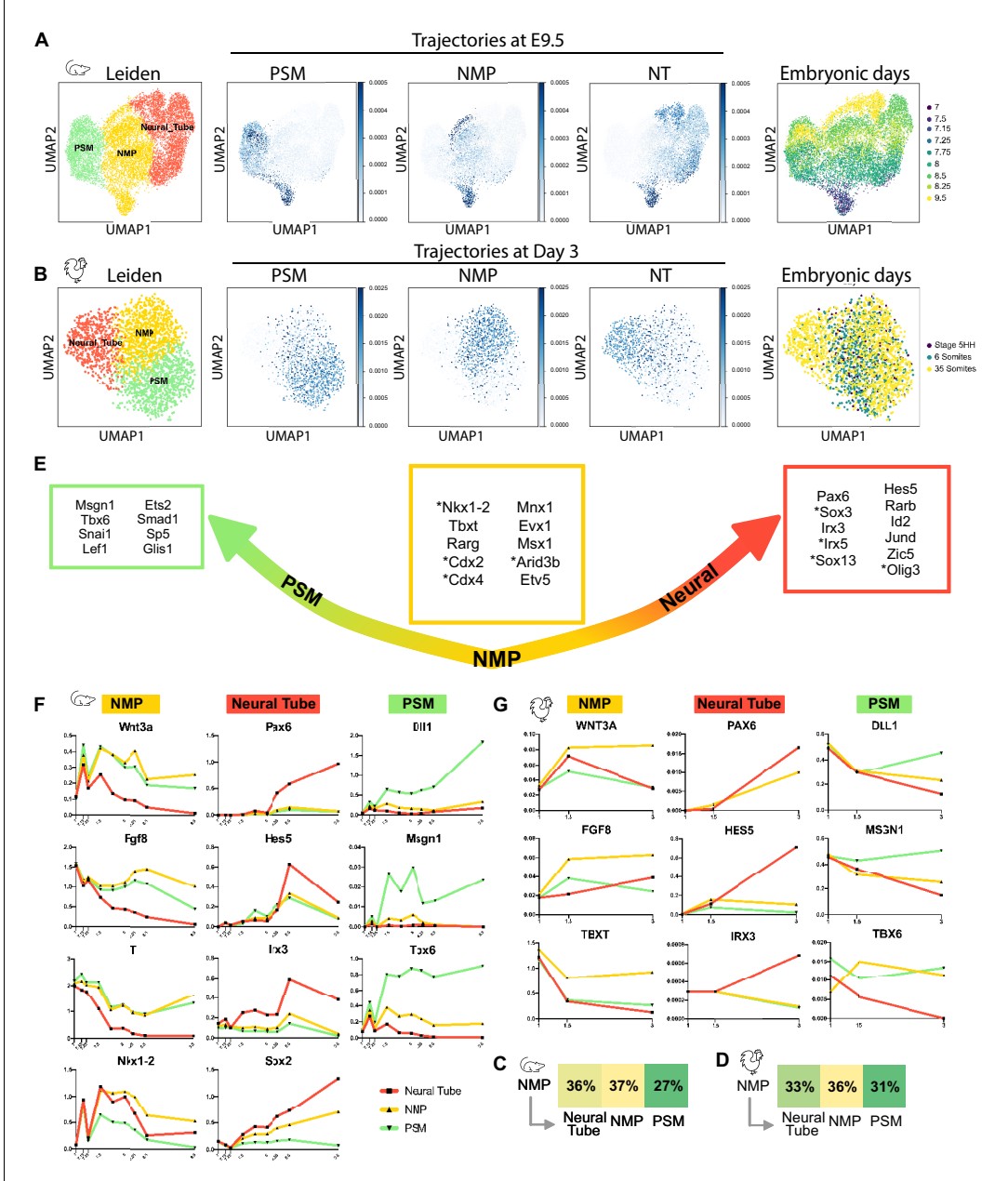

**Figure 4.** NMPs trajectory analysis in silico. (**A, B**) (left) Uniform Manifold Approximation and Projection (UMAP) projection of cells of the NMP, neural, and PSM clusters, extracted from the mouse (**A**) and chicken (**B**) datasets showing the cell types identified following Leiden unsupervised clustering and differential gene-expression analysis. 2362 cells for the chicken dataset and 12072 cells for the mouse dataset. (Middle) Developmental trajectories of the cells of the NMP, PSM, and neural tube clusters at day E9.5 in the mouse dataset and stage 35 somites in the chicken dataset identified with the Waddington-OT pipeline. Optimal transport was used to infer temporal couplings in the mouse dataset at time E7, E7.15, E7.25, E7.5, 7E.75, E8, E8.5, E9, and in the chicken dataset at stage 5HH, 6 somites, 35 somites, subclustered for NMP, PSM, and neural tube. (Right) Distribution of cells by developmental age. (**C, D**) Transition tables representing the amount of mass transported from NMPs to the other cell types from day E7 to day E9.5 for the mouse dataset (**C**) and from stage 5HH to 35 somites for the chicken dataset (**D**). (**E**) Predicted transcription factors enriched in cells most likely to transition to each particular fate from day E7 to day E9.5 for the mouse dataset and from stage 5HH to 35-somites for the chicken dataset (transcription factors found in the mouse dataset but not in the chicken dataset are annotated with *). (**F, G**) Normalized Log gene expression of the predicted transcription factors for each cell type during axis elongation in mice (**F**) and chicken (**G**). NMP: neuromesodermal progenitors; PSM: presomitic mesoderm.

The online version of this article includes the following figure supplement(s) for figure 4:

**Figure supplement 1.** Predictive transcription factor gene trends along trajectory of the neuromesodermal progenitor (NMP) fate.

also found in chicken NMPs as well as Nkx1.2*,Cdx2*, Cdx4*, Arid3b*, which were very low or not detected in the chicken dataset (*Figure 4E–G*, *Figure 4—figure supplement 1*). Plotting the expression level of these genes in the three clusters during development shows that they are all preferentially expressed in the NMP cluster (*Figure 4—figure supplement 1*). Interestingly, transcription factors such as Mnx1 have been identified in the chordoneural hinge of mouse embryos, where NMPs are proposed to reside (*Wymeersch et al., 2019*), but their role in NMP differentiation has not been investigated. Altogether, our single-cell analysis in mouse and chicken identified cells within the NMP clusters as ancestors of cells from both PSM and neural tube. It also identified interesting transcription factor candidates potentially implicated in maintenance and differentiation of NMPs.

## An epithelium to mesenchyme transition during NMP maturation

While in the combined datasets NMPs form a single cluster, we nevertheless observed a temporal trajectory within the NMP cluster in both species (*Figures 3J, M* and *4A, B*). Reanalyzing the NMP cells only with Leiden clustering identified NMP early and late clusters in both chicken and mouse embryo datasets (*Figure 5A–H*, *Supplementary files 1* and *2*). As expected, a major difference between these clusters is linked to the collinear activation of Hox genes with expression of paralogs 9–13 being restricted to the late clusters (*Figure 5D, H*). Removing Hox genes from the analysis still led to the identification of early and late clusters respectively characterized by sets of specific genes (*Figure 5—figure supplement 1A, B*, *Supplementary file 2*). Significant overlap was observed when comparing the top differentially expressed genes in the mouse and chicken early and late clusters to published early and late NMP gene lists (*Supplementary file 4; Dias et al., 2020*; *Diaz-Cuadros et al., 2020*; *Gouti et al., 2017*; *Wymeersch et al., 2019*). We noted that, in both species, genes associated with an epithelial state such as *EPCAM* or *CDH1* were significantly upregulated in the early cluster (as expected due to the epithelial nature of the epiblast at these stages) and downregulated in the late ones (*Figure 5—figure supplement 1E, F*, *Supplementary file 4*). In contrast, genes preferentially associated to a mesenchymal state such as *ZEB1, VIM,* or *MMP2* were upregulated in the late clusters (*Figure 5—figure supplement 1E, F*). To assess globally if NMP early and late cells were undergoing EMT, we performed a Gene Set Enrichment Analysis (GSEA) using the epithelial to mesenchymal transition gene set (*Subramanian et al., 2005*). We find a positive enrichment score (NES) only for the late NMP clusters in chicken and mouse, suggesting that NMP cells in the tail bud are undergoing Epithelial to Mesenchymal Transition (EMT) (*Figure 5I–L*). To confirm these observations, we performed immunostaining of the epiblast and tail bud sections from stage 5 and stage 18HH (30–36 somites) chicken embryos with the epithelial marker E-cadherin (*Figure 5M, N*). Consistent with our single-cell data analysis, we found that SOX2/T double-positive cells in the epiblast (i.e., early NMPs) exhibit strong apical expression of CDH1. In contrast, SOX2/T cells in the tail bud (i.e., late NMPs) do not express CDH1 while strong expression is detected in the adjacent epithelial ectoderm. This argues that during their maturation SOX2/T cells lose their original epithelial characteristics in the tail bud, concomitantly with expression of the more posterior Hox gene paralog groups.

## Comparison of NMP signature genes across species and datasets

We compared our top list of genes differentially expressed in NMPs identified in chicken (304 genes, false discovery rate (FDR) < 0.05) and mouse (top 350 genes, FDR < 0.05) datasets (*Supplementary file 5*). We found 44 conserved genes between the two species showing a striking enrichment in Wnt pathway genes including known targets such as *Axin2, T,* and *Fgf8* as well as pathway members including *Wnt3a/5a/5b/8a* and *Wls.* We next compared the signature genes for the chicken NMP population identified in this study with NMP signatures identified in mouse embryos (*Dias et al., 2020*; *Diaz-Cuadros et al., 2020*; *Gouti et al., 2017*; *Guibentif et al., 2021*) as well as in human SOX2/T-positive cells differentiated in vitro (*Diaz-Cuadros et al., 2020*; *Supplementary file 5*). We identified 38 genes conserved in four or five of the six datasets. This list includes signaling proteins such as Wnt5a, Wnt8a, Fgf17, and Cyp26a1 as well as T and Nkx1.2. Interestingly, the glucose transporters (Slc2a3 [glut3] and Slc2a1 [glut1]) as well as the lactate dehydrogenase isoforms (Ldha and Ldhb) were also identified in this list consistent with the high level of glycolysis occurring in the tail bud of chicken and mouse embryos (*Bulusu et al., 2017*;

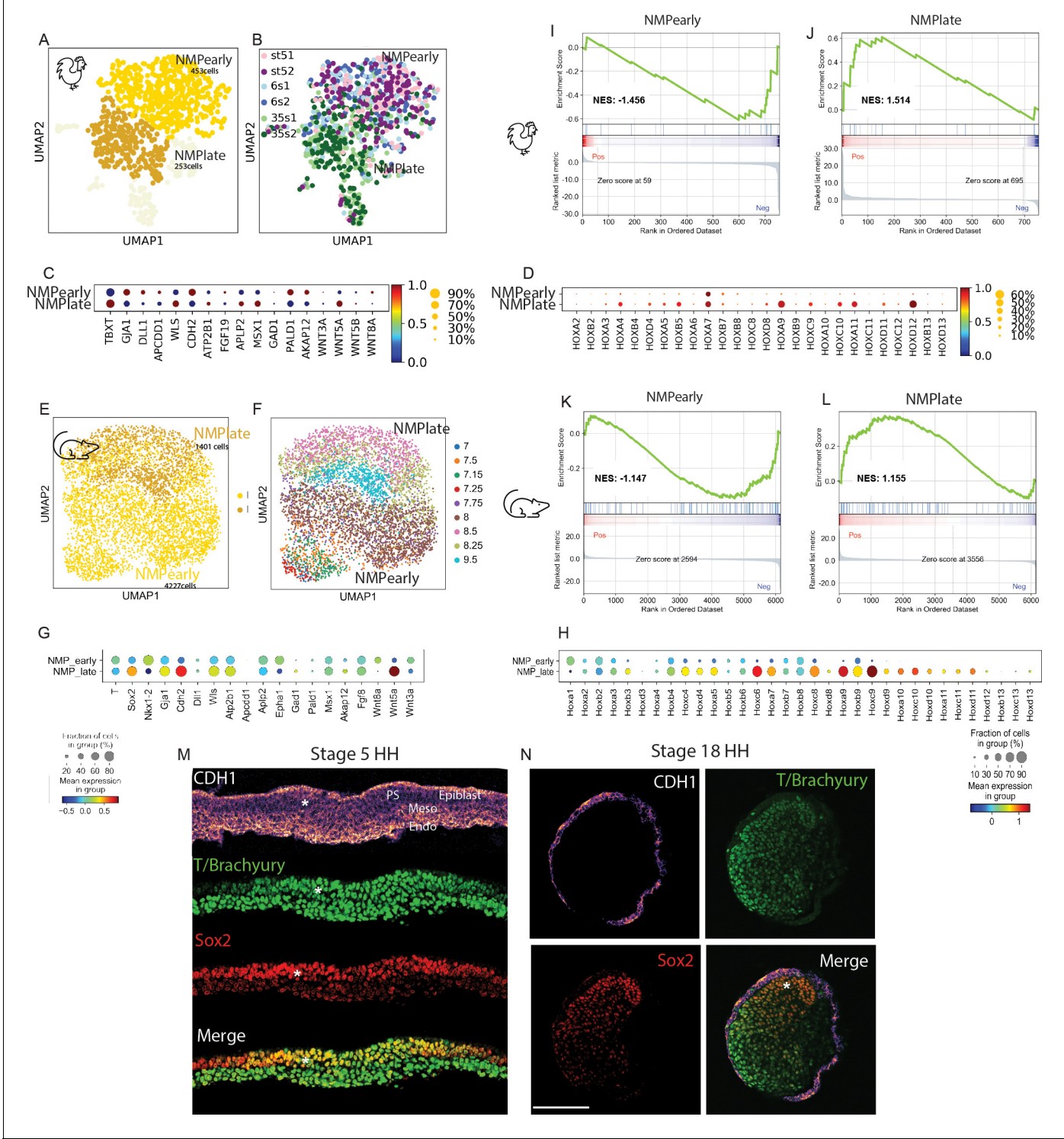

**Figure 5.** Downregulation of neuromesodermal progenitors' (NMPs) epithelial phenotype during development. (**A, B**) *k*-NN graphs showing cells of the chicken NMP cluster identified in the analysis shown in **Figure 3H** (total: 706 cells) from stage 5HH, 6-somite, and 35-somite visualized with Uniform Manifold Approximation and Projection (UMAP) and analyzed using Leiden clustering. Major clusters include an NMP early cluster (gold) and an NMP late cluster (goldenrod) (**A**). (**B**) Distribution of cells by developmental age. (**C, D**) Dotplot showing the expression levels of NMP signature (**C**) and HOX genes (**D**) in chicken NMP clusters. (**E, F**) *k*-NN graphs showing cells of the mouse NMP cluster identified in the analysis shown in **Figure 3K** (total: 5628 cells) from stage E7.0–E9.5, visualized with UMAP and analyzed using Leiden clustering. Major clusters include the early NMP and late NMP clusters (**E**). (**F**) Distribution of cells by developmental age. (**G**) Dotplot showing the expression levels of NMP signature genes in mouse NMP clusters. Note that

*Figure 5 continued on next page*

Figure 5 continued

*Fgf19* is not expressed in mouse. Both *Sox2* and *Nkx1.2* genes are detected in mouse data and added to the dotplot. (H) Dotplot showing the expression levels of Hox genes in mouse NMP clusters. (I–L) Gene Set Enrichment Analysis (GSEA) of early NMP clusters in chicken (I) and mouse (J) and of late NMP clusters in chicken (K) and mouse (L) using the Hallmark Epithelium to Mesenchymal Transition gene set. The normalized enrichment score (NES) is based on the gene set enrichment scores and accounts for differences in gene set size and in correlations between gene sets and the expression dataset. The top portion of the plot shows the running enrichment score (ES) for the gene set as the analysis walks down the ranked list. The middle portion of the plot shows where the members of the gene set appear in the ranked list of genes. The bottom portion of the plot shows the value of the ranking metric as you move down the list of ranked genes. (M, N) Representative immunostaining of E-cadherin/CDH1, T/Brachyury, and SOX2 in cryosections of the NMP-containing anterior PS region in stage 5HH (M) and of the tail bud region of stage 18HH (N) in chicken embryos. PS: primitive streak; Endo: endoderm; Meso: mesoderm; PSM: paraxial mesoderm; Nt: neural tube. Asterisk shows the NMP domain dorsal to the top (M). D: dorsal; V: ventral and anterior to the top (N). Scale bar: 100 µm (n = 3 embryos). Circle sizes in (C, D, G, H) show the percentage of cells expressing the gene in the cluster. Color shows the normalized level of expression. Normalization is done by clusters across all the Hox genes.

The online version of this article includes the following source data and figure supplement(s) for figure 5:

Source data 1. Time in the primitive streak before ingression from the tracking.

Figure supplement 1. Characterization of the early and late neuromesodermal progenitor (NMP) clusters.

*Oginuma et al., 2017*). This list also included genes coding for transcription factors such as Cdx1/2/4, Evx1, Hes3, Sp5/8, and Hoxc9. Other genes usually associated to NMPs such as Sox2, Wnt3a, Fgf8, an Axin2 were shared by three out of the six gene signature lists. When the chicken NMP signature identified in this study was compared to the transcriptional signature of human NMP-like cells (top 350 genes), we identified 30 conserved genes including WNT5B, WLS, T, SP5/8, AXIN2, APCDD1, EVX1, LDHB, and B3GT7 as conserved between the two species. These genes were also conserved when comparing our mouse NMP signature to the human SOX2/T cells dataset but also included genes such as NKX1.2, SLC2A3, and CDX1/2. (*Supplementary file 5*). Overall, our data argues for a conserved gene expression signature of NMP cells between chicken mouse and human, presenting a striking enrichment of genes of the Wnt and glycolytic pathways.

## Limited convergence and ingression of the NMP territory in the epiblast

We next analyzed the cellular dynamics in the epiblast during PS regression. We performed long-term tracking of epiblast cells from stage 4+HH to 15 somites after nuclear cell labeling using the live marker nuclear red (*Video 3*). We measured the longevity of epiblast cell tracks to localize zones where trajectories end due to cell ingression (*Figure 6A, B, Video 4*). Posterior cell trajectories are less persistent than anterior ones, indicating that cells in the posterior part of the PS exit the epiblast layer sooner than cells of the anterior region. The tracks of cells in the anterior PS region show mainly angles with the midline between 0° and 45°, indicating limited convergence toward the midline. In contrast, epiblast cells in the posterior half of the PS show angles from 45° to 90° throughout PS regression, suggesting that these cells converge toward the midline to join the PS (*Figure 6C, D, Figure 6—figure supplement 1*). To measure the speed of cell convergence along the PS, we plotted the instantaneous speed of cells in the lateromedial ($V_{LM}$) and anteroposterior ($V_{AP}$) directions over time as a function of their position along the PS (*Figure 6E*). This revealed a low $V_{LM}$ in the anterior PS region compared to the posterior PS. In contrast, epiblast cells of the anterior PS region show a high $V_{AP}$, similar to that of the node, suggesting that they follow the node posterior movements. $V_{AP}$ decreases progressively in the LP domain to become minimal in the posterior-most region. A similar analysis at a later stage of PS regression (stage 6 HH-5-somites) revealed similar cell dynamics and trajectories for epiblast cells of the anterior PS region maintaining low lateral to medial and high anteroposterior speed (*Figure 6—figure supplement 1B, C*). Thus, epiblast cells exhibit a posterior to anterior gradient of convergence speed toward the PS (*Figure 6F*).

To quantify the global dynamics of cell ingression along the PS, we selectively marked epiblast cells by applying nuclear red dorsally at stage 4+HH. This procedure, when performed in ovo, only labels the dorsal epiblast but not ingressed mesodermal cells or the endoderm. We performed confocal movies from the ventral side of the embryo to measure the increase of mean fluorescence intensity over time along the PS (*Figure 6G, Video 5*). Since only epiblastic cells are marked at the beginning of the movie, the mesodermal layer will progressively acquire new fluorescent cells over time, reflecting the dynamic of cell ingression. We observed a gradual posterior to anterior increase

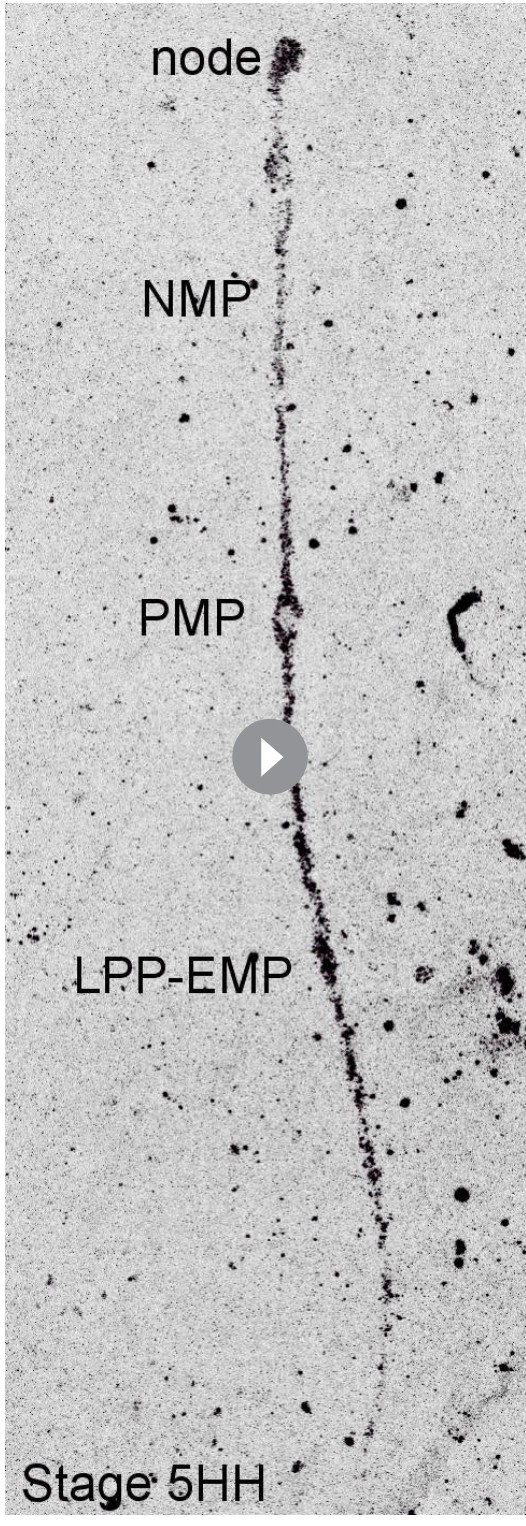

**Video 3.** Long-term tracking of a nuclear red stained embryo. Time-lapse movie of dorsal maximum z-projections of epiblast cells labeled with nuclear red showing their tracks from stage 5HH to 5 somites during PS regression. Colors indicate the z-position from magenta (dorsal) to yellow (ventral). t = 4 min between frames, z-sectioning = 4 μm. 20× objective, LSM 780. NMP: neuromesodermal progenitors; PMP:

of fluorescence over time in the mesoderm along the PS (*Figure 6G, H*). In addition, we could fit the different curves of intensity measurements along the anteroposterior axis to a linear curve, indicating an anteroposterior gradient of cell ingression of epiblastic cells along the PS (*Figure 6H*). This ingression pattern parallels the distribution of laminin along the PS, which progressively disappears posteriorly (*Figure 6I*), consistent with a more active ingression behavior. Together, these experiments demonstrate that the epiblast gradient of convergence speed is coupled to graded cell ingression along the PS anteroposterior axis with minimal ingression in the anterior PS region where NMP cells reside.

## An anterior to posterior gradient of proliferation counteracts ingression in the anterior PS epiblast

We noted that the number of SOX2/T cells gradually increases from stage 4+HH to reach a peak around 30 somites (*Figure 7A*). As limited convergence and ingression is observed in the NMP territory, this increase is most likely explained by cell proliferation. During PS regression, we observed a higher number of phospho-histone 3 (pH3)-positive cells relative to the total number of cells (mitotic index) in the anterior region of the PS compared to more posterior ones (*Figure 7B, C*). We performed confocal live imaging of fluorescent H2B-Cherry quail embryos and observed more dividing cells in the anterior PS compared to more posterior regions at the same developmental stage (*Figure 7D*). This confirmed the existence of a higher mitotic index in the anterior region compared to more posterior parts of the PS (*Stern, 1979*).

We next manually tracked individual electroporated cells and measured the time spent in the PS prior to cell ingression (*Figure 7E, F*). All the cells tracked in the mid PS region spent from 1 to 3 hr in the PS before ingressing. In contrast, only 35% of the tracked cells in the anterior PS region show such fast ingression dynamics (within 1–3 hr), whereas 65% remain in the PS for more than 7 hr (*Figure 7F*). The number of SOX2/T cells increases from 50 to 550 cells in around 40 hr (*Figure 7A*). Knowing the percentage of ingressing cells in the anterior PS region (35%), we can predict the evolution of the cell population using a geometric series formula: $U_n = q^n \times U_0$ classically used in analysis of population dynamics. Here, n is the number of cell divisions, q is the doubling parameter of the non-ingressed population (here 2 * 0.65 = 1.3), and $U_0$ is the initial population,

presomitic mesoderm progenitors; LPP: lateral plate progenitors. PS = primitive streak; NT: neural tube. https://elifesciences.org/articles/64819#video3

that is, 50 cells. To obtain a population of 550 cells, the model predicts n = 9 or 10 cell cycles ($U_9$ = 530, $U_{10}$ = 689), suggesting a cell cycle time around 4 hr. We manually tracked individual dividing cells and their daughter cells in the anterior PS region over 10 hr to determine the time between two divisions. We identified symmetric cell divisions where the two daughter cells remain in the epiblast after cell division, consistent with self-renewal of this population. The cell cycle time for such symmetric divisions is around 4.5 hr in agreement with the number of divisions predicted by the model (*Figure 7G*, *Figure 6—figure supplement 1A, B*, *Video 6*). Other symmetric cell divisions gave rise to two daughter cells entering the mesoderm (*Figure 6—figure supplement 1*). We also observed asymmetric cell divisions where one of the daughter cells ingresses after cell division, thus suggesting a specification of one of the daughter cells to a mesodermal fate (*Figure 6—figure supplement 1A, B*). Thus, we show that cells in the NMP region exhibit rapid cell divisions together with limited cell ingression, allowing their self-renewal and amplification during formation of the posterior body.

## Posterior to anterior exhaustion of PS progenitor territories results in NMPs remaining as the major PS remnant in the tail bud

The posterior gradients of convergence speed and ingression combined with increased proliferation in the anterior PS region are expected to lead to progressive asymmetric disappearance of the PS precursor territories in a posterior to anterior order. To test this hypothesis, we generated time-lapse movies of GFP-expressing transgenic chicken embryos from stage 5HH to 10-somite. We tracked specific positions along the PS approximately corresponding to the boundaries between the NMP and paraxial mesoderm progenitors (PMPs), the PMP and lateral plate progenitors (LPPs), and LPP and extraembryonic mesoderm progenitor (EMP) (*Figures 1D* and *7A, B*, *Video 7*). We observed a faster reduction of the posterior PS domains, with the extraembryonic territory disappearing first, followed by the LP territory whose ingression is completed at the 10-somite stage (*Figure 8A, D*; *Moreau et al., 2019*; *Spratt, 1947*). After this stage, most T-positive progenitors of the superficial layer of the tail bud also express SOX2, suggesting that they correspond to the remnant of the epiblast flanking the anterior PS and remain the only axial progenitors left in the tail bud (*Figure 8D*). Thus, the precursor territories along the PS do not disappear at the same rate. We observe a sequential posterior to anterior exhaustion of the territories of the extraembryonic mesoderm, the lateral plate, and the SOX2-negative PMPs, which results in finally locating the NMP territory in the tail bud (*Figure 8D*).

## Discussion

NMPs are a population of stem cells that generate most of the posterior spinal cord, vertebrae, and skeletal muscles. Surprisingly, this cell population was only recently discovered using retrospective clonal analysis in mouse (*Tzouanacou et al., 2009*). The identification of such an important population of stem cells came much as a surprise because none of the many classical fate mapping studies of the epiblast and PS of the chicken or mouse embryo performed so far ever reported the identification of such bipotent precursors (*Brown and Storey, 2000*; *Iimura et al., 2007*; *Psychoyos and Stern, 1996*; *Schoenwolf et al., 1992*; *Selleck and Stern, 1991*; *Tam and Beddington, 1987*; *Wilson and Beddington, 1996*; *Wilson et al., 2009*). While the retrospective technique employed to first identify mouse NMPs unambiguously identified these bipotent stem cells, it could however not locate them in the embryo (*Tzouanacou et al., 2009*). Thus, direct identification of the amniote NMPs and their location in the embryo is still lacking. Here, we used lineage tracing and scRNAseq to identify and characterize NMPs as a population of stem cells located in the SOX2/T-expressing region of the anterior PS epiblast in the chicken embryo. We show that cells expressing the mesodermal marker T and the neural marker SOX2 are first found in the epiblast adjacent to the anterior PS and Hensen's node region at the beginning of PS regression. This domain appears at a similar stage of mouse development and occupies a position similar to the SOX2/T domain of the node-streak border and the caudal lateral epiblast proposed to contain NMPs in mouse embryos (*Wymeersch et al., 2016*). In both mouse and chicken embryos, this domain contains cells fated to

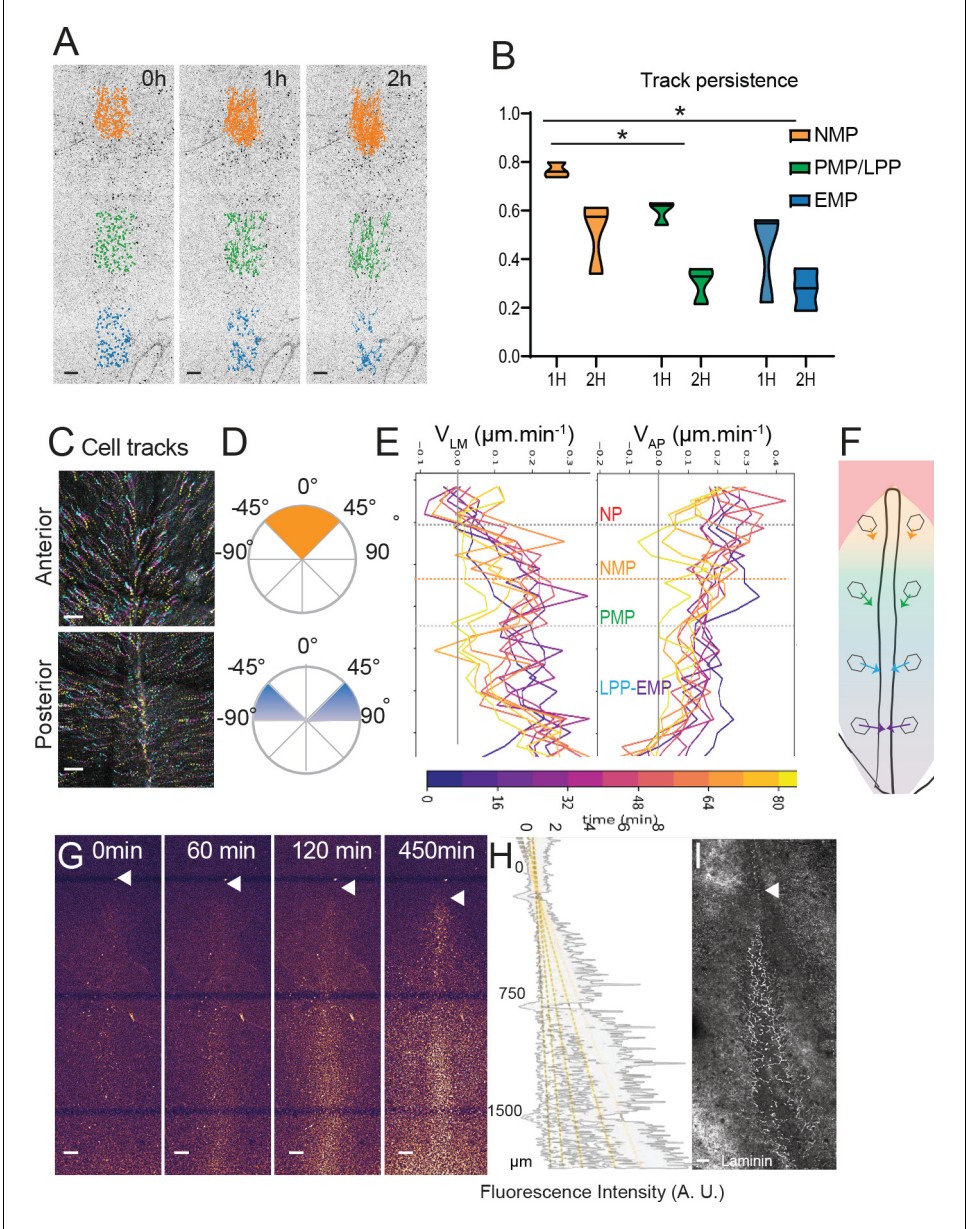

**Figure 6.** Limited convergence and ingression of the NMP territory in the epiblast. (**A**) Representative snapshots from a time-lapse movie of a stage 5HH chicken embryo in which the epiblast was labeled with nuclear red (**Video 4**). Tracks of single-labeled nuclei are shown at three different PS levels at three different timepoints (0, 1, and 2 hr) to illustrate differences in longevity of the cells (n = 3 embryos) (NMP: gold; PMP/LPP: green; EMP: blue). (**B**) Quantification of tracks longevity measured as the ratio of tracks number after 1 and 2 hr divided by the number of tracks at $t_0$ in each of the colored regions shown in **Figure 3A**. $t_0$ marks the start of time-lapse movies of stage 5HH chicken embryos labeled with nuclear red. Gold, green, and blue show tracks in the NMP, PMP/LPP, and EMP domains, respectively, corresponding to the tracks shown in **Figure 5A** (n = 3; n = 1502 tracks). Two-way ANOVA NMP-PMP/LPP; NMP-EMP. *p<0.05. (**C**) Representative color-coded time projection showing tracks of epiblast cells at the anterior and posterior PS level after nuclear red labeling in a stage 5HH chicken embryo. The tracks color code represents early timepoints in cyan and later timepoints in yellow (n = 3 embryos). (**D**) Representative quantification of the angle with the midline of tracks shown in (**C**). Top: anterior PS region; bottom: posterior PS region (n = 3 embryos). (**E**) Representative mean lateral to medial speed ($V_{LM}$) and anterior to posterior speed ($V_{AP}$) over time of epiblast cells labeled with nuclear red in stage 5HH chicken embryos. Y axis represents AP position along the embryo. Color code indicates time of measurement since beginning of the movie (n = 3 embryos). (**F**) Diagram showing the main direction of epiblast cell movements as a function of their AP position in the epiblast. The length of arrows is proportional to convergence speed. (**G**) Representative

*Figure 6 continued on next page*

*Figure 6 continued*

snapshots from a confocal movie of the PS region of a chicken embryo labeled dorsally at stage 5HH with nuclear red and imaged from the ventral side to show epiblast cells ingression (n = 3 embryos). (**H**) Representative intensity measurement of the nuclear red signal from the ventral side along the PS of the movie shown in (**G**). Y axis, distance to Hensen's node (n = 3 embryos). (**I**) Representative whole-mount immunohistochemistry with anti-laminin (white) in a stage 5HH chicken embryo. Ventral view (n = 3 embryos). Dorsal views, anterior to the top. EMP: extraembryonic progenitors; NMP: neuromesodermal progenitors; PMP: presomitic mesoderm progenitor; LPP: lateral plate progenitor; PS: primitive streak. Arrowhead shows Hensen's node position. Scale bar: 100 μm.

The online version of this article includes the following source data and figure supplement(s) for figure 6:

**Source data 1.** Double-positive T/SOX2 cells, pattern of cell division, and time in the primitive streak before ingression.

**Figure supplement 1.** Quantification of longevity, cell speed, and trajectories in time-lapse movies of nuclear red-labeled chicken embryos.

---

give rise to both neural and mesodermal derivatives, suggesting that they are functionally equivalent (*Garcia-Martinez et al., 1993*; *Wymeersch et al., 2016*). However, so far the bipotentiality of these cells has not been established at the single-cell level. We performed lineage tracing using a bar-coded retroviral library and Brainbow-derived MAGIC markers (*Loulier et al., 2014*) to show that single cells of the SOX2/T region are bipotential and can contribute both to the neural tube and paraxial mesoderm along the trunk axis in chicken embryos. We further demonstrate clonal continuity and transcriptional homogeneity between early NMPs of the PS and late ones in the tail bud.

The significant contribution of cells of the SOX2/T territory of the anterior PS to both neural and mesodermal lineages has been missed in previous fate mapping studies of this region in chicken embryos (*Brown and Storey, 2000*; *Fernández-Garre et al., 2002*; *Henrique et al., 1997*; *Iimura et al., 2007*; *Psychoyos and Stern, 1996*; *Schoenwolf et al., 1992*; *Selleck and Stern, 1991*). Compared to these studies, we analyzed our lineage-tracing experiments at significantly later stages after in ovo labeling (stage 17–20HH instead of stage 10–14HH). Thus, while these fate maps indicate that cells of the anterior PS epiblast initially produce either neural or mesoderm descendants, a trend also observed in mouse clones (*Tzouanacou et al., 2009*), NMPs can give rise to both lineages but mostly later, in more posterior regions of the body. These observations are consistent with recent grafts of the epiblast territory in 6-somite chicken embryos showing that the territory first produces neural and then both mesodermal and neural derivatives (*Kawachi et al., 2020*). Direct identification of bipotential cells with a neural and mesodermal fate has been reported in zebrafish where they segregate during gastrulation and contribute to the most posterior part of the axis (*Attardi et al., 2018*). In zebrafish, however, only monopotent cells were found in the tail bud (*Attardi et al., 2018*; *Kanki and Ho, 1997*), while this is not the case in chicken (this report) and mouse (*Tzouanacou et al., 2009*).

scRNAseq analysis of the posterior embryonic region during PS to tail bud transition in chicken and mouse revealed a distinct NMP cluster lying between a cluster of PSM cells and one of neural cells. Using two different trajectory inference analysis methods (diffusion pseudotime and WOT), we identified two developmental trajectories leading from NMPs to these neural and PSM clusters, consistent with the bipotentiality of these cells. While the mouse NMP cluster shows expression of the NMP markers *T, Sox2,* and *Nkx1.2*, only *T* was identified in the chicken NMP cluster. *SOX2* and *NKX1.2* (*SAX1*) expression was neither found in the NMP cluster nor in neural tissue, suggesting a problem with annotation in the chicken genome. Alternatively, this might reflect a difference in sequencing depth as the chicken dataset was obtained using the inDrops pipeline, whereas most of the mouse data was obtained with the 10X platform. From our analysis, we identified a signature of genes enriched in the NMP clusters, including a majority of effectors and targets of wnt signaling, which is known to play a key role in the differentiation of this lineage (*Henrique et al., 2015*). Most of these signature genes are expressed in all chicken and mouse NMP clusters. Using WOT trajectory analysis, we identified cells within the NMP cluster as ancestors of the paraxial mesoderm and neural clusters in the mouse and chicken embryo. We were not able to find any bias for the NMP contribution toward the neural or mesodermal fate. This result aligns with our lineage-tracing data, which does not show any preferential enrichment toward a specific fate in our analysis timeframe. Interestingly, we identified genes associated to the NMP trajectory, including previously undescribed

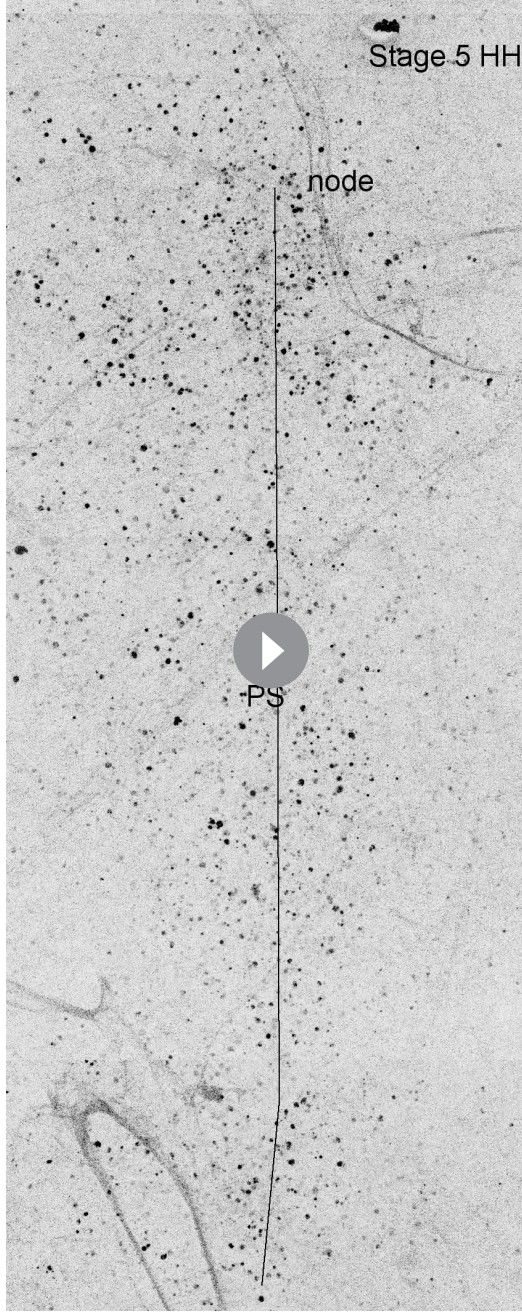

**Video 4.** Longevity of tracks along the primitive streak (PS). 3.5 hr time-lapse movie of a stage 5HH chicken embryo in which the epiblast was labeled with nuclear red. Tracks of single-labeled nuclei are shown at three different AP levels of the PS to illustrate differences in longevity of the cells. Dorsal view (neuromesodermal progenitor [NMP]: gold; paraxial mesoderm precursor [PMP]/lateral plate progenitor [LPP]: green; extraembryonic mesoderm progenitor [EMP]: blue). t = 4 min. 20× objective, LSM 780.
https://elifesciences.org/articles/64819#video4

ones such as RARG or MNX1 in both the mouse and chicken datasets. We also identified expected transcription factors that are predictive of the fate differentiation toward the neural (PAX6) and PSM lineages (MSGN1). It will be interesting to investigate how these genes function during differentiation of these lineages. Using diffusion pseudotime and WOT analyses, we also identified a developmental trajectory within the NMP cluster, suggesting that NMP cells undergo maturation during axis formation. This is supported by the identification of clusters of early and late NMPs characterized by expression of specific genes in both chicken and mouse. As reported for mouse embryos (*Wymeersch et al., 2019*), late NMPs are characterized by the expression of more posterior Hox genes in both chicken and mouse. We also observed that genes associated to the epithelial state such as *EpCAM, CDH1,* or *GJA1* are downregulated while genes associated to the mesenchymal state are upregulated at tail bud stages in both species. This is consistent with observations in mouse where tail bud progenitors were shown to undergo incomplete EMT during later stages of axial elongation (*Dias et al., 2020*).

As reported for mouse embryos (*Wymeersch et al., 2016*; *Wymeersch et al., 2019*), we observe an increase in SOX2/T cell numbers during axis elongation, indicating that these cells can self-renew while giving rise to a progeny in the paraxial mesoderm and in the neural tube. Labeling experiments in mouse and chicken embryos have identified a population of epiblast cells in the region of the anterior PS and Hensen's node, which behave as stem cells, giving rise to descendants in the paraxial mesoderm while being able to self-renew (*Cambray and Wilson, 2002*; *Cambray and Wilson, 2007*; *Iimura et al., 2007*; *McGrew et al., 2008*; *Nicolas et al., 1996*; *Selleck and Stern, 1991*; *Wilson et al., 2009*). These cells were proposed to contribute mostly to medial somites while lateral somitic cells are derived from more posterior areas of the PS (*Iimura et al., 2007*; *Selleck and Stern, 1991*). The anterior SOX2/T territory encompasses Hensen's node and the epiblast adjacent to the anterior PS and approximately corresponds to the territory containing the stem cells fated to give rise to medial somites. The epiblast territory immediately posterior to the SOX2/T territory does not express SOX2 and exhibits many cells positive for the paraxial mesoderm-specific marker MSGN1 (this study). This territory likely corresponds to the prospective territory of lateral somites, which does not show

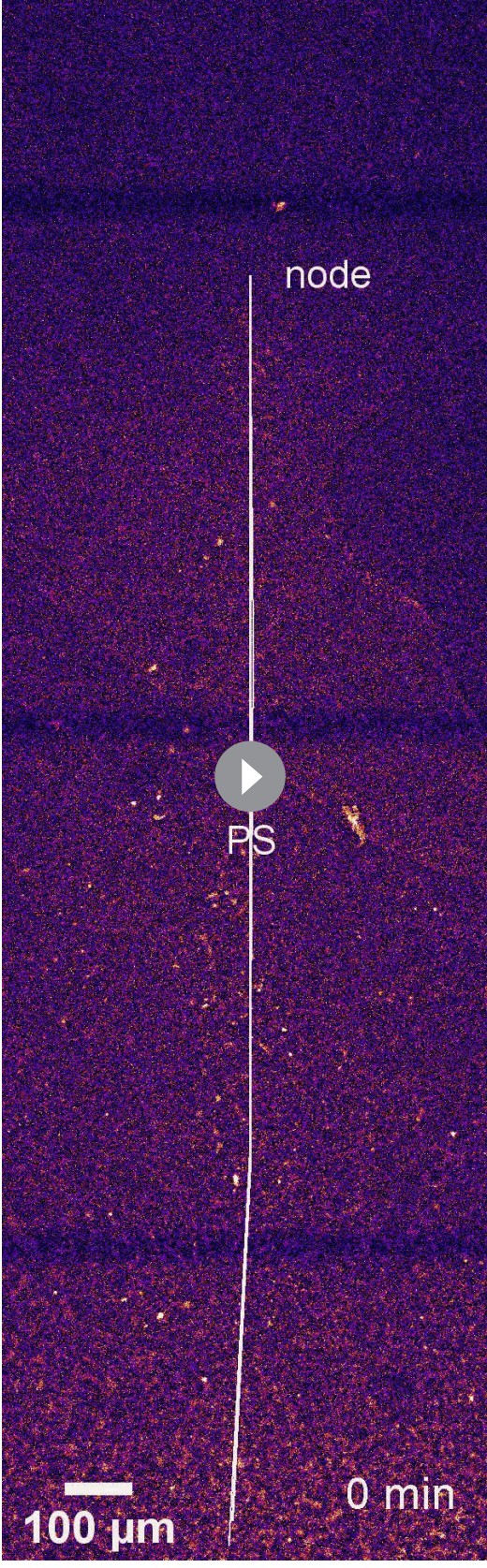

**Video 5.** Dynamics of mesodermal cell ingression. Time-lapse movie showing ventral z-maximum

a stem cell behavior, giving rise to descendants spanning only ~5–7 segments (*Iimura et al., 2007*; *Psychoyos and Stern, 1996*). Our data suggest that this territory becomes largely exhausted at the end of PS regression, resulting in posterior somites to derive mostly from the NMPs (*Figure 8D*). This is consistent with observations in chicken and mouse demonstrating that the selective contribution of different PS territories to medial or lateral somitic territories only applies to anterior somites (*Cambray and Wilson, 2007*; *Psychoyos and Stern, 1996*).

We observed a posterior to anterior gradient of convergence in the epiblast associated to a parallel gradient of ingression in the PS. A parallel posterior to anterior gradient of cell motility of ingressed mesodermal cells has been documented along the regressing PS (*Zamir et al., 2006*). These graded movements in the mesoderm are largely controlled by a gradient of Fgf8 acting as a repellent on newly ingressed mesodermal cells at this stage (*Yang et al., 2002*). This cellular dynamics suggests that the movement away from the posterior PS could act as a sink for the epiblastic territories generating these progenitors. These cell movements in the epiblast and the ingressing mesoderm could explain the progressive exhaustion of the PS from its posterior end first described for the extraembryonic territory (*Spratt, 1947*). Importantly, our data show that the precursor territories along the PS do not disappear at the same rate. We observe a sequential posterior to anterior exhaustion of the territories of the extraembryonic mesoderm, the lateral plate, and the SOX2-negative PMPs (*Figure 8D*). Combined to an increased proliferation in the NMP region, this could explain why the SOX2/T territory eventually remains as the major remnant of the PS in the tail bud after PS regression. Thus, while most of the PS behave as a transit zone for committed progenitors as proposed by *Pasteels, 1937*, its anterior-most region behaves more like a blastema, containing multipotential cells as predicted by Wetzel (*Romanoff, 1960*; *Wetzel, 1929*). The fact that the tail bud contains multipotent NMPs in addition to monopotent territories such as the precursors of the notochord or the hindgut could explain the long-standing controversy on whether the tail bud functions as a blastema or as a mosaic of committed precursors (*Catala et al., 1995*; *Davis and Kirschner, 2000*; *Gont et al., 1993*; *Holmdahl, 1925*).

Here, we show that NMPs can contribute to a single type of derivative (neural or mesodermal) for several segments followed by another type in

projections of the primitive streak (PS) region of a chicken embryo labeled dorsally at stage 5HH with nuclear red. t = 15 min. z = 4 μm. 20× objective, LSM 780.

https://elifesciences.org/articles/64819#video5

the next segments, as reported in mouse embryos (*Tzouanacou et al., 2009*). This argues for a striking plasticity of NMPs and rules out simple models of asymmetric divisions for the generation of neural and mesodermal descendants. Furthermore, as reported in mouse, we also observed bifated clones that do not contribute to the tail bud (*Tzouanacou et al., 2009*). Plasticity of the NMPs is supported by heterotopic grafts of the NMP epiblast territory into territories fated to become neural or meso-dermal, which resulted in the donor cells to adopt the fate of their new territory in chicken or mouse embryos (*Garcia-Martinez et al., 1997*; *McGrew et al., 2008*; *Wymeersch et al., 2016*). Our results also show that the transcriptional identity of the NMP population remains largely stable throughout axis formation. This suggests that the NMP population behaves as a unique stem cell population that remains uncommitted toward either lineage, with its descendants acquiring their identity only after entering the territory of the paraxial mesoderm or the neural tube. Such plasticity is supported by the observation that SOX2/T-positive NMP-like cells generated in vitro can be induced to differ-entiate to either neural or mesodermal cells by changing the culture conditions (*Diaz-Cuadros et al., 2020*; *Edri et al., 2019a*; *Edri et al., 2019b*; *Gouti et al., 2017*; *Gouti et al., 2014*; *Henrique et al., 2015*; *Turner et al., 2014*). The difficulty to observe NMPs in vivo contrasts with the ease with which NMP-like cells can be obtained in vitro from mouse or human pluripotent stem cells. NMPs have been generated in 2D cultures by activation of Wnt signaling at the epiblast-like stage (*Diaz-Cuadros et al., 2020*; *Edri et al., 2019a*; *Edri et al., 2019b*; *Gouti et al., 2017*; *Gouti et al., 2014*; *Henrique et al., 2015*; *Turner et al., 2014*). Analysis of 3D cultures such as gastruloids induced in vitro from mouse ES cells also shows that most of the cells forming these structures belong to the neural tube and paraxial mesoderm lineage, suggesting that they are largely derived from an initial NMP population (*Beccari et al., 2018*; *Faustino Martins et al., 2020*; *van den Brink et al., 2020*). NMP-like cells generated in vitro express both SOX2 and T and single-cell RNA-sequencing demon-strated that they form a very homogeneous population with characteristics similar to the endoge-nous SOX2/T cells (*Diaz-Cuadros et al., 2020*; *Gouti et al., 2017*). Our comparison of chicken, mouse, and human NMP cells identified by scRNAseq points to an significant level of conservation of the transcriptional signature of this population. Overall, our work answers an important conun-drum on the existence and fate of the NMPs in vivo, an elusive stem cell population in amniotes.

# Materials and methods

## Key resources table

| Reagent type (species) or resource | Designation | Source or reference | Identifiers | Additional information |
|---|---|---|---|---|
| Strain, strain background | *Gallus gallus* | Charles River Laboratories, RRID:SCR_003792 | Specific-pathogen-free chicken (SPF) eggs | |
| Strain, strain background | *Gallus gallus* | Susan Chapman at Clemson University; South Carolina; USA, RRID:SCR_011159 *McGrew et al., 2004* | | Cytoplasmic GFP eggs |
| Strain, strain background | Northern Bobwhite quail | Ozark Hatcheries *Bénazéraf et al., 2017* | | Transgenic quails expressing H2B-Cherry |
| Chemical compound, drug | Paraformaldehyde | Sigma | 158127 | |
| Antibody | T/Brachyury (goat polyclonal) | R&D Systems | AF2085 | IF (1/1000) |
| Antibody | SOX2 (rabbit polyclonal) | Millipore RRID:AB_2286686 | Cat# AB5603 | IF (1/1000) |

*Continued on next page*

*Continued*

| Reagent type (species) or resource | Designation | Source or reference | Identifiers | Additional information |
|---|---|---|---|---|
| Antibody | E-Cadherin (mouse monoclonal) | Abcam RRID:AB_1310159 | ab76055 | IF (1/250) |
| Antibody | N-Cadherin (rabbit polyclonal) | Abcam RRID:AB_298943 | ab12221 | IF (1/250) |
| Antibody | Fibronectin (mouse) | DSHB | MT4S | IF (1/50) |
| Antibody | Laminin (rabbit) | Sigma-Aldrich RRID:AB_477163 | Cat# L9393 | IF (1/200) |
| Antibody | Phospho-histone 3 (rabbit polyclonal) | Santa Cruz Biotechnology RRID:AB_2233067 | sc-8656 | IF (1/1000) |
| Antibody | MSGN1 (rabbit polyclonal) | Pourquie Laboratory *Oginuma et al., 2017* | | IF (1/1000) |
| Recombinant DNA reagent | pCAGG-H2B- Venus | Pourquie Laboratory *Denans et al., 2015* | | pCAGG backbone |
| Recombinant DNA reagent | pCAGG-H2B-RFP | Pourquie Laboratory *Denans et al., 2015* | | pCAGG backbone |
| Recombinant DNA reagent | pCAGG-GAP43-Venus | Pourquie Laboratory *Oginuma et al., 2017* | | pCAGG backbone |
| Chemical compound, drug | NucRed Live 647 ReadyProbes Reagent | Thermo Fisher | R37106 | Two drops in 1 ml |
| Software, algorithm | MOSAIC plug-in from ImageJ | ImageJ *Sbalzarini and Koumoutsakos, 2005* | | |
| Software, algorithm | Tracs reconstruction | Arthur Michaut | This study | Python-based homemade code |
| Software, algorithm | K mean Brainbow clustering | *Figure 2—source data 2* | MATLAB | |
| Recombinant DNA reagent | pQCGICIDPA | Retroviral barcoding | This study | pQCXIX backbone |
| Recombinant DNA reagent | Tol2-CAG::Nucbow | RRID:Addgene_158992 *Loulier et al., 2014* | | pCX backbone |

## Chicken embryos

All animal experiments were performed in accordance to all relevant guidelines and regulations. The office for protection from Research Risks (OPRR) has interpreted 'live vertebrate animal' to apply to avians (e.g., chick embryos) only after hatching. All of the studies proposed in this project only concern early developmental stages (prior to 5 days of incubation); therefore, no IACUC-approved protocol is required. Fertilized chicken eggs were obtained from commercial sources. Fertilized eggs from transgenic chickens expressing cytoplasmic GFP ubiquitously (*McGrew et al., 2004*) were obtained from Susan Chapman at Clemson University. Fertilized eggs from transgenic quails expressing H2B-Cherry (*Bénazéraf et al., 2017*) were obtained from Ozark Hatcheries. Eggs were incubated at 38℃ in a humidified incubator, and embryos were staged according to *Hamburger and Hamilton, 1992*. We cultured chicken embryos mainly from stage 5HH on a ring of Whatman paper on agar plates as described in the EC culture protocol (*Chapman et al., 2001*).

## Immunohistochemistry

For whole-mount immunohistochemistry, stage 3–20HH chicken embryos were fixed in 4% paraformaldehyde (PFA; 158127, Sigma) diluted in PBS 1X at 4℃ overnight. The embryos were rinsed and permeabilized in PBS-0.1% Titon, three times 30 min, and incubated in blocking solution (PBS-0.1% Triton, 1% donkey serum; D9663, Sigma) prior to incubating with primary and secondary antibodies. Embryos were incubated in antibodies against T/Brachyury (1/1000, R&D Systems: AF2085), SOX2 (1/1000, Millipore: ab5603), E-cadherin (1/250 Abcam: ab76055), N-cadherin (1/250, Abcam: ab12221), fibronectin (1/50 DSHB: MT4S), laminin (1/200, Sigma: L9393), phospho-histone 3

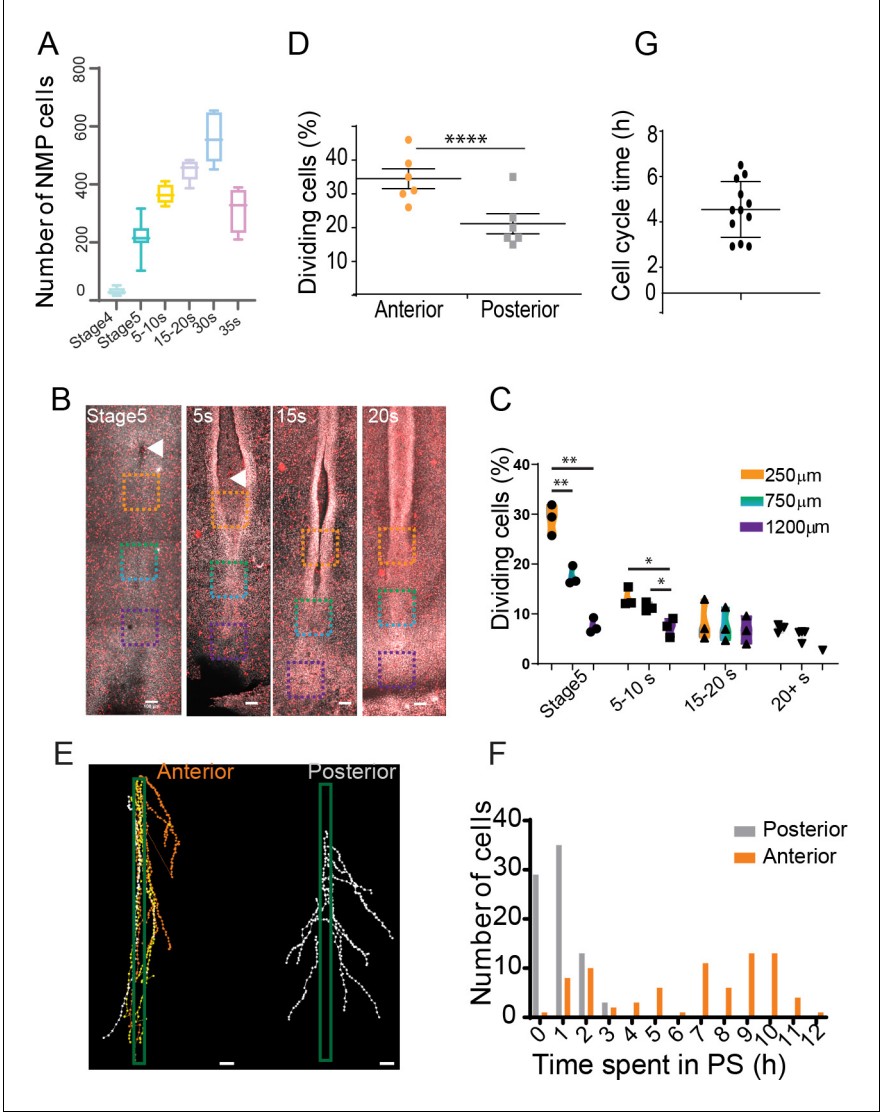

**Figure 7.** An anterior to posterior gradient of proliferation counteracts ingression in the anterior primitive streak (PS) epiblast. (**A**) Quantification of the number of SOX2/T double-positive cells in chicken embryos from stage 4HH to 35-somite (n = 38 embryos). (**B**) Snapshots of the posterior region of chicken embryos from stage 5HH to 20-somite stained in whole-mount with an anti-phosphorylated histone H3 (pH3) antibody. (**C**) Quantifications of the mitotic index along the PS in the boxes shown in (**B**). Orange box: 250 µm from node; green box: 750 µm from node; blue box: 1200 µm from node (n = 13 embryos). Unpaired t-test; **p=0.0017 and 0.0022; *p=0.0187, p=0.0278. (**D**) Quantification of the number of dividing cells in H2B-Cherry transgenic quails at stage 4 +/5HH in the anterior and posterior PS (n = 6). Paired t-test; ***p=0.0001. (**E**) Tracks and (**F**) quantification of trajectories of the neuromesodermal progenitor (NMP) (gold) and lateral plate progenitor (LPP) (gray) cells during PS regression (n = 159 cells, 80 posterior, 79 anterior in seven embryos). (**G**) Quantification of the time interval between two rounds of division in cells of the NMP region measured in time-lapse movies (n = 12 inter-division events in four embryos). (**B**, **E**) Dorsal views, anterior to the top. Arrowhead shows Hensen's node position. Scale bar: 100 µm.

The online version of this article includes the following source data and figure supplement(s) for figure 7:

**Figure supplement 1.** Analysis of cell division profiles in the SOX2/T region.

**Figure supplement 1—source data 1.** Type of cell division in the neuromesodermal progenitor (NMP) domain.

(1/1000, Santa Cruz: sc-8656), MSGN1 (1/1000) (*Oginuma et al., 2017*), diluted in blocking solution at 4°C overnight. Embryos were rinsed and washed three times 30 min in PBS-0.1% Triton, incubated 1 hr in blocking solution and incubated at 4°C overnight with secondary antibodies conjugated with

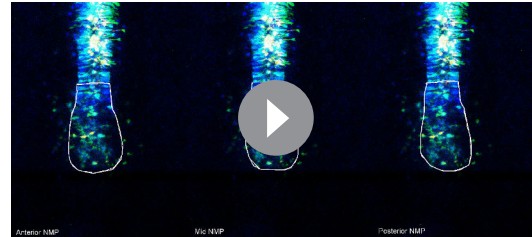

**Video 6.** Tracking cell division time in the SOX2/T region. Tracking of cell divisions in cells of the SOX2/T region electroporated with an H2B-RFP plasmid at stage 5HH. d1, d2, d3: division events; PS: primitive streak. t = 6 min. 10× objective, Leica DMR.
https://elifesciences.org/articles/64819#video6

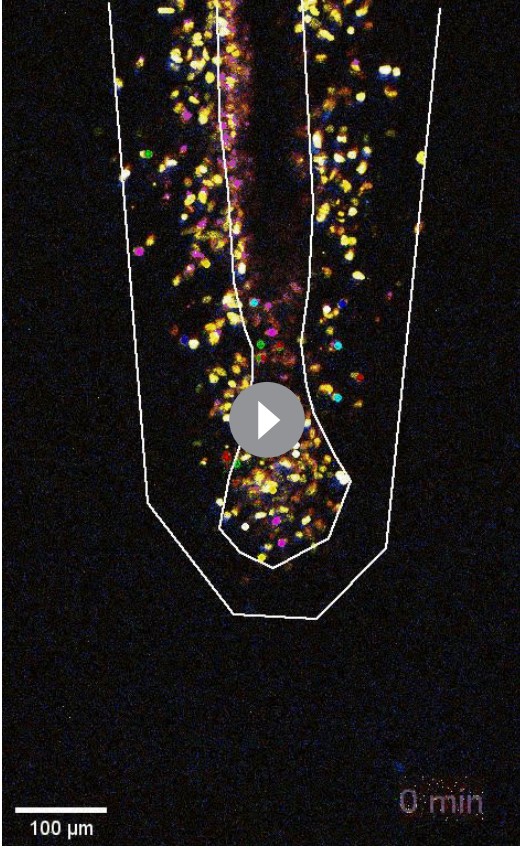

**Video 7.** Long-term tracking of the mesodermal progenitors. 13 hr time-lapse movie of a GFP chicken embryo starting at stage 5HH (left). Fate of the color-coded mesodermal primitive streak (PS) progenitor territories during regression is shown in the right movie. NMP: neuromesodermal progenitors (gold); PMP: presomitic mesoderm progenitors (green); LPP: lateral plate progenitor; LPP-EMP: extraembryonic mesoderm precursors (blue-purple). Arrowhead: Hensen's node. Anterior to the top. t = 4 min. 20× objective, LSM 780.
https://elifesciences.org/articles/64819#video7

Alexa Fluor (Molecular Probes) diluted in blocking solution. If the staining was not imaged in the following two days, post-fixing was performed using a 4% PFA solution.

For histological analysis, stage 5HH chicken embryos were fixed in 4% PFA. Embryos were then embedded in OCT compound and frozen in liquid nitrogen. Frozen sections (12 µm) were cut using a Leica Cryostat and incubated overnight at 4°C with the primary antibody diluted in blocking solution (same as above), and after washing in PBS-0.1% Triton, they were incubated overnight at 4°C with the secondary antibody conjugated with Alexa Fluor (Molecular probes) diluted in blocking solution. Sections were then washed in PBS-0.1% Triton before mounting in fluoromount-G (Thermo Fisher) and stored at 4°C overnight prior to imaging.

Images were captured using a laser scanning confocal microscope with a 10× or 20× objective (LSM 780, Zeiss). To image the whole embryo, we used the tiling and stitching function of the microscope (5 by 2 matrix) and z-sectioning (5 µm). Later stages (from 17HH) were imaged in clearing solution using the scale A2 clearing protocol from *Hama et al., 2011*. For imaging, the embryo was placed in the clearing solution (4 M urea 0.1% Triton, 10% glycerol) 30 min prior to imaging in glass bottom dishes (Mattek).

## Plasmid preparation and electroporation

pCAGG-H2B-Venus and pCAGG-H2B-RFP have been described in *Denans et al., 2015*, and pCAGG-GAP43-Venus has been described in *Oginuma et al., 2017*. Chicken embryos at stage 4 or 5HH were prepared for in ovo electroporation. Eggs were windowed and a DNA solution (1 µg/µl) mixed in PBS, 30% glucose, and 0.1% Fast Green was microinjected in the egg, in the space between the vitelline membrane and the epiblast in the first 500 µm posterior to the node or 1500 µm posterior to the node to target the NMP or the LPPs, respectively. Electroporation was carried out using two pulses at 5 V for 1 ms on each side of the PS in the NMP and LP domains (four locations) using a needle electrode (CUY614, Nepa Gene, Japan) and an ECM 830 electroporator (BTX Harvard Apparatus). This procedure only labels the superficial epiblast layer (*Iimura and Pourquié,*

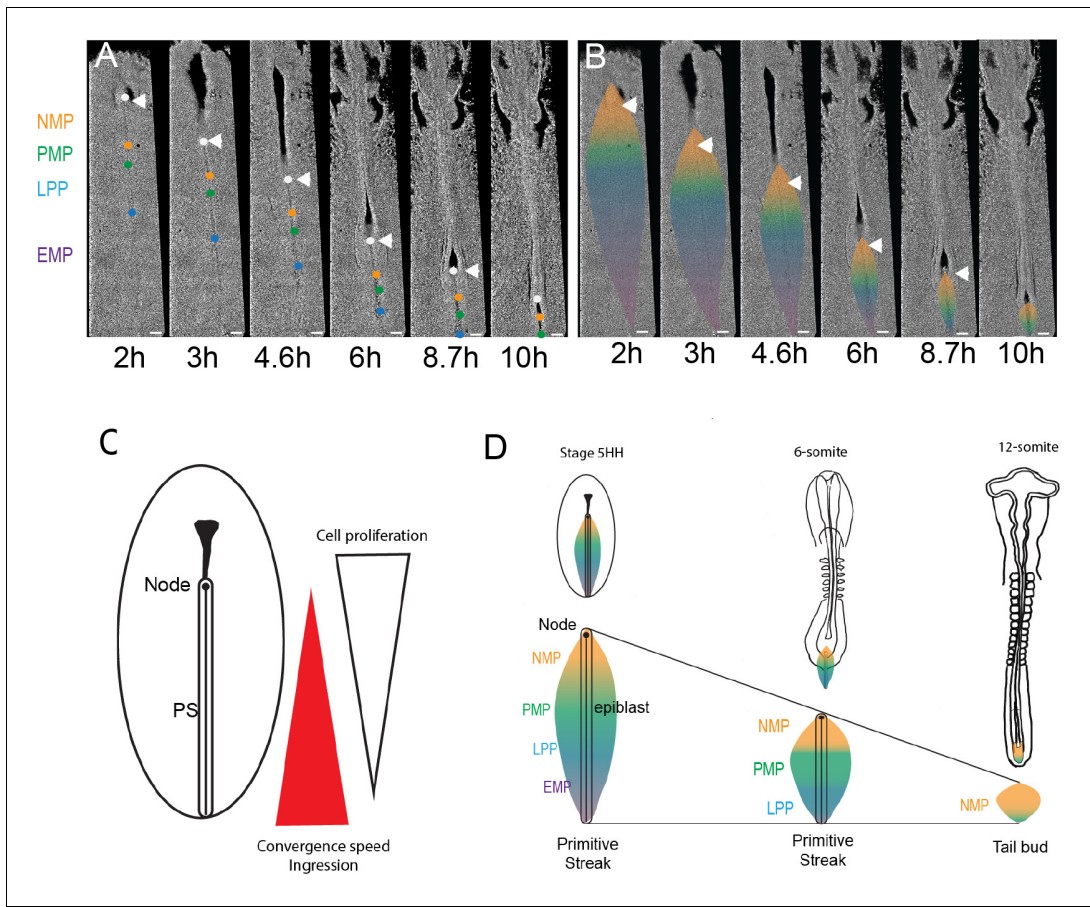

**Figure 8.** Posterior to anterior exhaustion of primitive streak (PS) progenitor territories results in NMPs remaining as the major PS remnant in the tail bud. (**A, B**) Snapshots from a 13 hr time-lapse movie of a chicken embryo starting at stage 5HH. The approximate position of the boundaries is shown by colored dots (**A**), and the corresponding territories are shown (**B**) during PS regression. Boundaries between NMP-PMP, PMP-LPP, and LPP-EMP are illustrated by orange, green, and blue dots (**A**) and color transition (**B**), respectively. The initial position of groups of cells marking the boundaries between the different PS territories was identified based on their distance to Hensen's node (white arrowhead) as established in our experiments and in *Psychoyos and Stern, 1996*. These groups of cells were tracked during PS regression to follow the fate of the different PS territories (n = 3 embryos). Dorsal views. (**C**) Schematics illustrating the position of the opposite gradients of cell proliferation and ingression in the epiblast during PS regression. (**D**) Schematics summarizing the dynamics of the NMP territory (gold) during PS regression in embryos of stage 5HH and 6-somite and around PS to tail bud transition (12-somite embryo). (**A–D**) Dorsal views, anterior to the top. AP: anteroposterior; NMP: neuromesodermal progenitors (gold); PMP: presomitic mesoderm progenitors (green); LPP: lateral plate progenitor (blue); EMP: extraembryonic mesoderm precursors (purple). Arrowhead: Hensen's node (n = 3 embryos). Scale bar: 100 μm.

*2008*). Eggs were then re-incubated for 1–2 hr at 38°C, and embryos were dissected and prepared for EC culture imaging in six-well imaging plates as described in *Denans et al., 2015*. For lineage-tracing experiments, we performed the same procedure as above but using only one pulse on each side of the PS in the NMP domain to minimize the number of electroporated cells.

## Nuclear red labeling in vivo

The nuclear red solution was prepared from the NucRed Live 647 ReadyProbes Reagent (Thermo Fisher) and diluted in PBS 1X as indicated by the manufacturer's experimental procedures. Sparse nuclear labeling of the dorsal epiblast was performed in ovo by injecting the nuclear red solution between the epiblast and the vitelline membrane at the PS level. The solution was left for 15 min to perform the long-term epiblast tracking and 30 min for monitoring ingression. The embryos were then dissected, rinsed in PBS, and mounted on paper filter for EC culture to perform live

imaging from the dorsal side for the long-term epiblast tracking and from the ventral side for measuring cell ingression.

Time-lapse imaging, ingression, and cell-cycle length measurements stage 5HH chicken embryos were cultured ventral side up on a microscope stage using a custom-built time-lapse station (*Bénazéraf et al., 2010*). We used a computer-controlled, wide-field (10× objective) epifluorescent microscope (Leica DMR) workstation, equipped with a motorized stage and cooled digital camera (QImaging Retiga 1300i), to acquire 12-bit grayscale intensity images (492 × 652 pixels). For each embryo, several images corresponding to different focal planes and different fields were captured at each single timepoint (frame). The acquisition rate used was 10 frames per hour (6 min between frames). To quantify cell ingression and division, the image sequence was registered to the node displacement by tracking the advancement of Hensen's node as a function of time using the manual tracking plug-in in ImageJ (*Denans et al., 2015*). Cell division events were manually tracked in long-term movies of H2B:GFP and RFP electroporated embryos. The time between divisions was estimated by counting the number of frames.

To analyze the dynamics of mesodermal precursor territories during PS regression, we tracked three small regions of the epiblast adjacent to the PS located at a distance of 500, 700, and 1000 μm from Hensen's node at stage 5HH in transgenic chicken embryos expressing GFP. Embryos were imaged with a Zeiss LSM 780 confocal microscope with temperature control. At stage 5HH, the entire PS is around 2000 μm in length. According to our fate mapping (see *Figure 1*) and *Psychoyos and Stern, 1996*, these three regions correspond to transition zones between the NMP-PSM, PSM-LP, and LP-EM territories, respectively. The NMP territory is maintained over the first 500 μm from the node and remains in the *sinus rhomboidalis* region during PS regression. The PSM progenitor domain is found posterior to the NMP territory in the anterior third of the PS approximately extending 670 μm posterior to the node. We observed that electroporation of the region located 750 μm posterior to the node is labeling LP cells (*Figure 3*). Thus, we tracked the area located 700 μm posterior to the node, which approximately corresponds to the transition zone between the precursors of paraxial mesoderm and LP. The posterior half of the PS in stage 5HH chicken embryos mostly gives rise to extraembryonic mesoderm (*Psychoyos and Stern, 1996*). Thus, we considered the boundary between lateral plate and EMPs to be at 1000 μm from Hensen's node. These different regions were followed using the manual tracking module from imageJ in 10 hr time-lapse movies spanning from stage 5HH to the 10-somite stage. *Video 7* shows a representative movie where the different territories have been color-coded to visualize their dynamics during PS regression.

## Cell tracking analysis

Whole epiblast cell tracking was performed using the MOSAIC plug-in from ImageJ (*Sbalzarini and Koumoutsakos, 2005*). Tracks were then visualized and analyzed using a custom code in Python. Cell velocities were computed by calculating the discrete displacements. In order to back-track cells in regions of interest, groups of cells were selected at any given time using selection tools provided by the Scikit-image package (*scikit-image contributors et al., 2014*). Trajectories were then plotted using the Matplotlib package (*Hunter, 2007*).

Color-coded time projections were generated using the color-coded projection module from Zen software. Early to late timepoints are shown using the following color sequence: cyan, magenta, and yellow.

## Cell proliferation analysis

Stage 5, 8, 12, 13HH chicken embryos were dissected, fixed in 4% PFA, and immunostained with phospho-histone H3 (pH3) antibody (1/1000, Millipore) as described above. In the penultimate wash, Hoechst (1/1000) was added to label cell nuclei. The number of proliferative cells was measured in 250 by 250 μm squares at three different locations along the PS (center of the area being at 250, 750, and 1200 μm from the node). Proliferative cells were identified by automated segmentation of pH3-positive cells using ImageJ via thresholding the images and using the analyze particle module. Hoechst-positive cells were also segmented using the same procedure and their number was used to normalize to the cell density in each square. The percentage of dividing cells was obtained by dividing the number of pH3 cells by the number of Hoechst-positive cells in each square and multiplying by 100. A minimum of five embryos was analyzed for each stage. We also performed similar

measurements on live H2B:Cherry quail embryo at stage 5HH in the anterior (around the node) and posterior (LPP region). We used a high threshold, which identifies cells in mitosis (that show a brighter Cherry signal due to chromatin condensation), while all the other cells in the tissue are identified with a lower threshold.

## Analysis of the localization and number of NMPs

Chicken embryos from stage 4 to 20HH were harvested and fixed in 4% PFA at 4°C overnight. Embryos were immunostained in whole mount using T/Brachyury, SOX2, and Alexa Fluor 647 Phalloidin (1/500 Thermo Fisher: A22287) combinations and imaged by confocal laser microscopy (LSM 780, Zeiss). SOX2/T-double positive cells were segmented and counted using ImageJ software. To identify the double-positive cells, cells in the T and SOX2 channel were manually thresholded (around 1.3% of the histogram). We used the image calculator tool from ImageJ to generate a new image by performing the image operation T AND SOX2 and identify the double-positive cells. Cell were then counted automatically by the 'Analyze Particle' module on ImageJ (particle size < 500 pixel square). Embryos were grouped by stage, and data is shown in *Figure 5A*. n = 7 at stage 4HH; n = 11 at stage 5HH; n = 6 at 5–10 somites; n = 5 at 15–20 somites; n = 5 at 30 somites; n = 4 at 35 somites and older. Total n = 38 embryos.

## Lineage tracing and quantification

### Cloning of a high-complexity barcoded retroviral genomic plasmid library

The plasmid library used to generate the barcoded retrovirus was based on the pQCXIX backbone. First, this backbone was digested with NotI and XhoI. A Gibson assembly reaction was performed using three PCR-generated fragments to insert the EGFP, IRES, and Cre recombinase, creating pQCGIC (*Gibson et al., 2009*). These PCR fragments were generated using the indicated primers from pCAG-EGFP, pQCXIX, and pCAG-Cre plasmids, respectively. This plasmid was then digested with XhoI and PvuII, and an intron disrupted polyadenylation sequence (IDPA) amplified from a gBlock (IDT) was inserted with a Gibson assembly reaction, creating pQCGICIDPA. Finally, the dsDNA barcode insert sequence was prepared from an Ultramer Oligo (IDT), QCGICbc_oligo, with PCR. This ultramer was synthesized with 12 SW repeats, resulting in a GC balanced 24 bp barcode. pQCGICIDPA was digested with XhoI, and a 170 µl Gibson assembly reaction was performed with 8.5 µg of linearized vector and 1.24 µg of insert. This reaction was purified and electroporated into three 100 µl aliquots of electrocompetent DH10β *Escherichia coli*. Following a 1 hr outgrowth in 12 ml SOC at 37°C with shaking, dilutions were plated on LB agarose plates with carbenicillin to determine complexity, and the remaining transformation was added to 500 ml of LB with carbenicillin and grown for 16 hr at 37°C. The liquid culture was split into four and plasmid was purified with the Qiagen Maxiprep Kit. A plate with $1/1 \times 10^4$ of the transformation grew approximately $1 \times 10^3$ colonies, indicating a total library complexity of $1 \times 10^7$.

| Target | Primer name | F primer sequence | R primer sequence |
|---|---|---|---|
| EGFP | CAGtoQCprel | CAGGAATTGATCCGCATTCTGCAGTCGACGGTACC | GAGGCCTACCGGTGCAAGTCAGATGCTCAAGGGGC |
| IRES | QCIRES | GCACCGGTAGGCCTC | GACGCGTGATCAAGCTTATCATC |
| Cre | CretoQCpostl | GCTTGATCACGCGTCGCAAAGAATTCTGAGCCGCC | TGGACCACTGATATCTCGAG CGGCCGCTATCACAGATCTT |
| IDPA | QCGICidpa | AGATCTGTGATAGCGGCCGCTCGAGATATCAG TGGTCCAGGCTCAATAAAGGGCAGGTAAGTATCAAGGTTAC | ATGGCTCGTACTCTATAGGC TTCAGACGCGTTCGGATTTGATC |
| QCGICbc_insert | QCGICbc | AGATCTGTGATAGCGGCCGCCCC TGGCTCACAAATACCACTGAGATCT | GAGCCTGGACCACTGATATCCCCCA TAATTTTTGGCAGAGGGAAAAAGATCT |
| IDPA gblock | | AATAAAGGGCAGGTAAGTATCAAGGTTACAAGACAGGTTTAAGGAGACCAATAGAAACTGGGCTTGTCG AGACAGAGAAGACTCTTGCGTTTCTGATAGGCACCTATTGGTCTTACTGACAT CCACTTTGCCTTTCTCTCCACAGGTGTCCACTCCCAGTTCTGTGTGTTGGTTTTTTGTGTGTTCTGGATCAAATCCGAACGCGT | |
| QCGICbc_oligo | | AGATCTGTGATAGCGGCCGCCCCTGGCTCACAAATACCACTGAGATCTSWSWS WSWSWSWSWSWSWSWSWSWAGATCTTTTTCCCTCTGCCAAAAATTATGGGGGATATCAGTGGTCCAGGCTC | |

## Preparation of replication-incompetent retrovirus

Replication-incompetent retrovirus was produced and concentrated as previously described (*Beier et al., 2011*; *Cepko and Pear, 2001*). Briefly, on day 1, eleven 150 cm plates with HEK 293T cells at 70–80% confluency were transiently transfected each with 13.5 µg of pQCGICbcIDPA, 6.75 µg pMN gagpol (*Ory et al., 1996*), and 2.25 µg of a plasmid expressing the EnvA envelope (*Landau and Littman, 1992*) using PEI (Polysciences, 24765; 90 µg per plate). Media was replaced on day 2, and viral supernatant was collected on days 3 and 4 and stored at −80℃. The supernatant was thawed, and virus was concentrated via ultracentrifugation (49,000 ×g for 2 hr at 4℃). Virus was resuspended in 270 µl of media and aliquoted in 10–20 µl aliquots and stored at −80℃. Titer was determined by infecting HEK 293T cells expressing the avian TVA receptor and was found to be approximately $4 \times 10^8$ infectious particles per milliliter.

## Retroviral infection

 1 µl of the virus preparation was diluted in 10 µl of PBS 1X and 0.1% Fast Green to form the virus mix. The virus mix was placed in ovo on top of the NMP region for 1 hr and washed with PBS 1X. The eggs where then sealed with tape and reincubated for 36 hr.

## Quantification

To retrieve the different barcodes, we manually harvested individual fluorescent cells from transverse sections of embryos fixed 36 hr post infection. GFP-positive cells were handpicked under a microscope (20× objective) using single-use needles (BD Precision). Each cell/needle was put in an individual PCR tube containing the buffer for the PCR1, and its localization was recorded. The tissue of origin of the cells was visually identified on the sections. The barcodes were retrieved using two consecutive nested PCR amplifications and sequenced using Sanger sequencing. PCR primer 1: 1R-TCTCTGTCTCGACAAGCCCAG, 1F-GATCATGCAAGCTGGTGGCTG. PCR primer 2: 2R-CTTACCTGCCCTTTATTGAGCCTG, 2F-CTGCTGGAAGATGGCGATGG.

## Nucbow cell tracing

Lineage tracing was performed by co-electroporating the NMP region at stage 5HH in ovo as described above with the following constructs: a self-excising Cre recombinase (se-Cre), the Nucbow construct, and the TolII transposase as described in *Loulier et al., 2014* in a 1/1/1 ratio at (1 µg/µl, each). We used similar concentrations for the Nucbow and transposase plasmids to that described in *Loulier et al., 2014* but increased 10 times the concentration (1 µg/µl versus 0.1 µg/µl) of the se-Cre to favor fast recombination and integration. Because non-integrated Nucbow plasmids can remain episomal and transiently affect the color of a cell, we performed our analyses after 36 hr when the plasmids are expected to have fully diluted through cell division. 36 hr after electroporation, we see that the number of fluorescent cells has significantly decreased, suggesting that the episomal transgenes have now been diluted.

To perform lineage analysis, we fixed the electroporated embryos at stage 17HH and 20HH, and imaged them in clearing solution ScaleA2 (*Hama et al., 2011*). The imaging was performed using an LSM 880 with Airyscan module in the three fluorescent channels following the same gating as in *Loulier et al., 2014*.

## Quantification

Cells were manually segmented in the YFP and Cherry channel using ImageJ. Positions were assigned to the mesodermal and neural tube layers. Color retrieval was performed by measuring the intensity in the three channels, Cerulean, YFP, and Cherry so that the total of all the intensities was normalized to 1 and expressed in percentages similarly to *Loulier et al., 2014*. Cluster assignment was performed using K-mean clustering followed by thresholding of only the cells with a silhouette >0.4. The coordinates were then calculated in a triplot diagram for visualization. The cells in a clone were then reassigned to their original localization based on the Y value as a proxy for the anteroposterior position in the image.

## scRNAseq analysis

### Preparation of single-cell suspensions for scRNAseq

Single-cell dissociation protocols were optimized to achieve >90% viability and minimize doublets before sample collection. To generate the samples, four embryos were harvested for each stage and cells were dissociated and captured on an inDrops (*Klein et al., 2015*) setup on the same day. For stage 5HH and 6-somite samples, the anterior half of the PS including Hensen's node and the posterior region of the neural plate were dissected. Two tail bud regions and two posterior ends (including the last somite formed, the PSM, and posterior neural tube and the tail bud) were dissected to generate the 35-somite sample. For single-cell dissociation, the dissected tissue was briefly rinsed in cold PBS and incubated in Accutase (Gibco) for 10–25 min at 37°C followed by mechanical dissociation. The cell suspension was analyzed with a hemocytometer to assess the quality of the dissociation and evaluate cell density. Dissociated cells were centrifuged at 350 g for 5 min at 4°C and resuspended at a concentration of 250,000 cells per microliter in 0.25% BSA in PBS. 2 × 3000 cells were sequenced per sample. Two biological replicates were collected per sample, and the sequencing data from both samples were combined for data analysis.

### Barcoding, sequencing, and mapping of single-cell transcriptomes

Single-cell transcriptomes were barcoded using the inDrops pipeline using V3 sequencing adapters as previously reported (*Klein et al., 2015*). Following within-droplet reverse transcription, emulsions consisting of about 3000 cells were broken, frozen at −80°C, and prepared as individual RNA-seq libraries. inDrops libraries were sequenced on an Illumina NextSeq 500 using the NextSeq 75 High Output Kits using standard Illumina sequencing primers and 61 cycles for read 1 and 14 cycles for read 2, eight cycles each for index read 1 and index read 2. Raw sequencing data (FASTQ files) were processed using the inDrops.py bioinformatics pipeline available at https://github.com/indrops/indrops, (copy archived at swh:1:rev:2ad4669b72ea6ba794f962ed95b778560bdfde4a); *Veres, 2021*. Transcriptome libraries were mapped to *Gallus gallus* transcriptome built from the GRCg6a (GCA_000002315.5) genome assembly. Bowtie version 1.1.1 was used with parameter –e 200.

### Processing of scRNAseq data

Single-cell counts matrices were processed and analyzed using ScanPy (1.4.3) and custom Python scripts (Code Availability). Low-complexity cell barcodes, which can arise from droplets that lack a cell but contain background RNA, were filtered in two ways. First, inDrops data were initially filtered to only include transcript counts originating from abundantly sampled cell barcodes. This determination was performed by inspecting a weighted histogram of unique molecular identifier–gene pair counts for each cell barcode and manually thresholding to include the largest mode of the distribution. Second, low-complexity transcriptomes were filtered out by excluding cell barcodes associated with <400 expressed genes. Transcript unique molecular identifier counts for each biological sample were then reported as a transcript × cell table, adjusted by a total-count normalization, log-normalized, batch corrected (using bbknn module), and scaled to unit variance and zero mean. Note that for the WOT analysis the module 'combat' was also used for batch correction. Unless otherwise noted, each dataset was subset to the 1000 most highly variable genes, as determined by a bin-normalized overdispersion metric.

### Low-dimensional embedding and clustering

Processed single-cell data were projected into a 50-dimensional PCA subspace ($k = 10$ except 35-somite $k = 15$) nearest-neighbor graph using Euclidean distance and 50 PCA dimensions and visualized using UMAP representation. Clustering was performed using Leiden community detection algorithms.

### Identification of differentially expressed genes

Transcripts with significant cluster-specific enrichment were identified by t-test comparing cells of each cluster to cells from all other clusters in the same dataset. Genes were considered differentially expressed if they met the following criteria: log-transformed fold change >0, adjusted p value<0.05. FDR correction for multiple hypothesis testing was performed as described by Benjamini–Hochberg. The top 100 differentially expressed genes, ranked by FDR-adjusted p values, associated fold

changes, and sample sizes (number of cells per cluster) are reported in *Supplementary file 1*. Gene names for the top 100 differentially expressed transcripts are reported in *Supplementary file 2*.

## Pseudo-spatiotemporal ordering and identification of differentially expressed genes

Pseudo-spatiotemporal orderings were constructed by randomly selecting a root cell from the NMP cluster and calculating the diffusion pseudotime distance of all remaining cells relative to the root. Trajectories were assembled for paths through specified clusters, with cells ordered by diffusion pseudotime values, as described in *Diaz-Cuadros et al., 2020*. Dynamically variable genes along the chicken NMP trajectory were identified as follows. In brief, sliding windows of 100 cells were first scanned to identify the two windows with maximum and minimum average expression levels for all genes individually. For each gene, a *t*-test was then performed between these two sets of 100 expression measurements (FDR < 0.01). Scaled expression values for significant genes were then smoothened over a sliding window of 100 cells, ranked by peak expression and plotted as a heat map, shown in *Figure 3—figure supplement 2D* and *Figure 3—figure supplement 3E*. The same method was used to plot the dynamically variable genes to identify each cluster type as shown in *Figure 3—figure supplement 1A–C*, *Figure 3—figure supplement 3E*, and the genes diagnostic of the epithelial mesenchymal transition in the NMP early and late clusters in *Figure 5—figure supplement 1E, F*.

## Machine-learning classification of cell states

Cell state prediction using the chicken cell states was predicted using the LDA classifier trained on the mouse E9.5 embryos (*Diaz-Cuadros et al., 2020*) after subsetting matching gene symbols for the E9.5 variable gene list and projecting into the E9.5-defined PCA subspace.

## Trajectory inference analysis using optimal-transport analysis

Waddington-OT 1.0.7 conceptual framework (*Schiebinger et al., 2019*) was used to infer the temporal couplings of cells from the different samples collected independently at various timepoints. Transport matrices (also called 'transport maps') were created by connecting each pair of timepoints and using an estimate of cellular growth rates. To estimate the growth rate, we scored each cell according to its expression of various gene signatures (proliferation and apoptosis) as described in *Schiebinger et al., 2019* and then model cellular growth with a Birth-Death Process, which assigns each cell a rate of division and a rate of death. The trajectory of a cell set refers to the sequence of ancestor distributions at earlier timepoints and descendant distributions at later timepoints. Ancestors were calculated by pushing back through the transport map and represented by intensity in the UMAP embedding of the dataset containing all cell sets. Transition tables were calculated to show the amount of mass transported from a cell type to another from a start and an end point. Predictive transcriptions factors for each cell fate of the datasets were obtained by creating fate matrix and searching for transcription factors that are enriched in cells most fated to transition to each particular fate (only transcriptions factor with a FDR < 0.01 and a fraction expressed ratio >1 between the timepoints were kept). Gene trends along trajectories were computed with non-scaled, log/normalized counts along trajectories.

## Gene Set Enrichment Analysis (GSEA)

List of genes and log fold-change values for each clusters obtained by differential expression (Wilcoxon rank-sum) were used as input for computing overlap with the Hallmark Epithelial Mesenchymal Transition gene set collection using the GSEA tool (*Liberzon et al., 2011*; *Subramanian et al., 2005*; *Subramanian et al., 1995*) (http://www.gsea-msigdb.org), and only genes with p-values<0.05 were kept. Results are plotted in *Figure 5* using the calculated normalized expression ratio (NES).

## **Data and code availability**

scRNAseq data and code are available on GitHub: https://github.com/PourquieLab/Guillot_2021 (copy archived at swh:1:rev:3d80f422bc5a675c08546cbb10ab79211e430b1e; *Pourquie Lab, 2021*).

## Acknowledgements

We thank members of the Pourquié lab, Domingos Henrique, Connie Cepko, Denis Duboule, Allon Klein, Dan Wagner, and Cliff Tabin for discussions and critical reading of the manuscript. We also thank Jean Livet for the MAGIC markers plasmids. We thank Dan Wagner, Samuel Wolock, and Jyoti Rao for their assistance with the sc-RNA analysis pipeline installation and discussions. Research in the Pourquié lab was funded by a grant from the National Institute of Health (RO1HD097068-02) and EMBO ALTF 406-2015 to CG. We thank the NeuroTechnology Studio at Brigham and Women's Hospital for providing the confocal LSM 880 instrument access and consultation on data acquisition and data analysis. We thank the single cell core at Harvard Medical School and the Bauer Core facility at Harvard University for their assistance with the single cell experiment and sequencing.

## Additional information

### Funding

| Funder | Grant reference number | Author |
| --- | --- | --- |
| EMBO | ALTF 406-2015 | Charlene Guillot |
| National Institutes of Health | RO1HD097068-02 | Olivier Pourquié |

The funders had no role in study design, data collection and interpretation, or the decision to submit the work for publication.

### Author contributions

Charlene Guillot, Conceptualization, Resources, Data curation, Software, Formal analysis, Supervision, Funding acquisition, Validation, Investigation, Visualization, Methodology, Writing - original draft, Project administration, Writing - review and editing; Yannis Djeffal, Formal analysis, YD performed the trajectory analysis in silico with CG and OP; Arthur Michaut, Software, AM wrote the code to analyze the cell trajectories and their speed; Brian Rabe, Resources, BR constructed the viral library and prepared the viral solutions; Olivier Pourquié, Supervision, Funding acquisition, Writing - original draft, Project administration, Writing - review and editing

### Author ORCIDs

Charlene Guillot (iD) https://orcid.org/0000-0001-9214-7509

### Decision letter and Author response

Decision letter https://doi.org/10.7554/eLife.64819.sa1
Author response https://doi.org/10.7554/eLife.64819.sa2

## Additional files

### Supplementary files

• Supplementary file 1. Table showing the cluster full names, cluster name abbreviation, and cell number at stage 5HH, 6 somites, 35 somites, all chicken embryo data, data subset from the chick (corresponding to *Figures 3H* and *5H*), chick and mouse data subsets shown in *Figures 4A, B* and *5E*, and the mouse subset data (*Figure 3K*).

• Supplementary file 2. Tables showing the differentially expressed genes per clusters at stage 5HH, 6 somites, 35 somites, all chicken data, chicken data subset, neuromesodermal progenitor (NMP) early and late reclustering with and without HOX genes and the mouse subsets. Each cluster is divided into three subcolumns where the first column indicates the gene name, the second column the p value, and the third the log fold changes. All the tables are ranked by z-score. The expression of the 10 first genes with the highest log fold change for each cluster is shown in the dotplots.

• Supplementary file 3. Predicted transcription factors identified by the Waddington-OT (WOT) analysis along the neuromesodermal progenitor (NMP), presomitic mesoderm (PSM), and neural trajectories. List of all the genes that are differentially expressed in the cells with the highest probability to

transition to each cell type; the list is then filtered by a curated list of transcription factors. The fraction expressed ratio (second column) represents the variation of cell fraction that express each gene between the first and second timepoint used to infer the trajectories (days 1–3 for chicken data and E7.25–E9.5 for mouse data). The false discovery rate (FDR) column show the FDR (adjusted p-value) for each gene.

• Supplementary file 4. Comparative analysis of the differentially expressed genes (DEGs) in the early and late neuromesodermal progenitor (NMP) clusters in mouse, chicken, and human models. We used the list of differentially expressed genes identified in *Dias et al., 2020*, *Guibentif et al., 2021*; *Gouti et al., 2017*; the top 350 genes of Table 5 from *Guibentif et al., 2021*. For the human gene list, we computed the list of DEG in the NMP cluster compared to the pluripotent stem cell cluster and took the 350 top genes from the *Diaz-Cuadros et al., 2020* study. For the list identified in this study, the genes that were kept are those that have a false discovery rate (FDR) > 0.05. To maintain a comparable number of genes, we decided to take only the top 350 genes when more than 1000 genes had an FDR > 0.05. In this table, we also compared the newly identified gene from this study to the list of up- and downregulated genes from *Wymeersch et al., 2019* providing from dissection of the node streak border NSB at E8.5 and the chordo-neural hinge CNH at E10.5 in mouse. The comparison of the different lists was done by using http://www.molbiotools.com/listcompare. For each analysis, we provide the matrix with the number of genes that are shared between two gene lists. We also provide the list of the shared items between all the lists and the gene list that intersect between two gene lists.

• Supplementary file 5. Comparative analysis of the neuromesodermal progenitor (NMP) gene lists in mouse, chicken, and human models. We used the list of differentially expressed genes (DEGs) identified in *Dias et al., 2020*, *Guibentif et al., 2021*; *Gouti et al., 2017*; the top 350 genes of Table 5 from *Guibentif et al., 2021*. For the human gene list, we computed the list of DEG in the NMP cluster compared to the pluripotent stem cell cluster and took the 350 top genes from the *Diaz-Cuadros et al., 2020* study. For the list identified in this study, the genes that were kept are those that have a false discovery rate (FDR) > 0.05. To maintain a comparable number of genes, we decided to take only the top 350 genes when more than 1000 genes had an FDR > 0.05 The comparison table was done by using http://www.molbiotools.com/listcompare. For each analysis, we provide the matrix with the number of genes that are shared between two gene lists. We also give the list of the shared items between all the lists and the gene list that intersect between two gene lists.

• Transparent reporting form

## Data availability

Sequencing data have been deposited in GEO under accession codes GSE161905. All the raw data to generate the graphs are given in a txt file.

The following dataset was generated:

| Author(s) | Year | Dataset title | Dataset URL | Database and Identifier |
|---|---|---|---|---|
| Guillot C, Pourquie O | 2019 | Single cell sequencing of posterior axis development in *Gallus gallus* | https://www.ncbi.nlm.nih.gov/geo/query/acc.cgi?acc=GSE161905 | NCBI Gene Expression Omnibus, GSE161905 |

The following previously published datasets were used:

| Author(s) | Year | Dataset title | Dataset URL | Database and Identifier |
|---|---|---|---|---|
| Pijuan-Sala | 2017 | Timecourse single-cell RNAseq of whole mouse embryos harvested between days 6.5 and 8.5 of development | https://www.ebi.ac.uk/arrayexpress/experiments/E-MTAB-6967/ | ArrayExpress, E-MTAB-6967 |
| Diaz-Cuadros | 2020 | In vitro characterization of the human segmentation clock | https://www.ncbi.nlm.nih.gov/geo/query/acc.cgi?acc=GSE114186 | NCBI Gene Expression Omnibus, GSE114186 |

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
