## [Decision Letter]

**Acceptance summary:**

This exciting paper characterizes the cellular dynamics of a population of stem cells, recently identified, called neuro-mesodermal progenitors in the chick embryo. By conducting prospective lineage trace and transcriptomic analysis, they identify dynamic cell behaviors that explain previous 'snapshot' studies showing that neuro-mesodermal progenitors contribute over long axial distances. This manuscript makes an important contribution to the understanding of how neuro-mesodermal progenitors contribute to vertebrate axial elongation and provides novel insights into axial morphogenesis.

**Decision letter after peer review:**

Thank you for submitting your article "Dynamics of primitive streak regression controls the fate of neuro-mesodermal progenitor cells in the chicken embryo" for consideration by *eLife*. Your article has been reviewed by 2 peer reviewers, and the evaluation has been overseen by a Reviewing Editor and Marianne Bronner as the Senior Editor. The following individual involved in review of your submission has agreed to reveal their identity: Valerie Wilson (Reviewer #2).

Essential Revisions

1. Many of the concerns can be addressed by text changes and minor changes to the existing data presentation as detailed in the full reviews.

2. The main problems are with the bioinformatic analysis, where both reviewers highlight issues where the analysis is too superficial and needs further work.

3. Additional imaging quantification is required to support their findings.

*Reviewer #1:*

The paper entitled "Dynamics of primitive streak regression controls the fate of neuro-mesodermal progenitor cells in the chicken embryo" from Guillot and colleagues aims to characterize in the context of developing chick embryos the cellular dynamics of a population of stem cells, recently identified, called Neuro-Mesodermal Progenitors (NMPs). The authors provide several lines of evidence supporting their claims that overall look interesting, however, additional imaging quantification and more comprehensive analysis of their transcriptomic datasets are required to support their findings.

Specifically, the authors generated an important scRNA-seq dataset of micro-dissect regions of the chick embryos across developmental time points from stage 5 to 35-somites. Despite the dataset represents a crucial part of the paper supporting the authors' findings of this NMP population, the analysis is not deep enough and does not report several controls and markers. In figure 2 the authors should show heatmap of the marker genes used for classification. Louvain markers genes that define each cluster should be showed in a supplementary not as an unreadable table, instead, a dotplot or heatmap would be better. It is unclear whether Louvain clustering has been used to label the data, did the authors find that each Louvain cluster correspond to a defined cell identity? The pseudotime analysis is incomplete and superficial. Diffusion pseudotime strongly depends on the root cell selection. RNA velocity analysis and/or another trajectory analysis are required to better show lineage trajectory. Figure 4 L-K, the pesudotime time is not progressively ordered and it would be advisable to perform an alternative analysis to visualize lineage trajectory. Furthermore, gene expression trends should be identified and further discussed over those trajectories. The supporting information about the classification analysis using the mouse dataset is incomplete.

– In figure 1 the authors show *SOX2*-T double staining. Panels A-B-C represent the first evidence of *SOX2*-T co-expression in stage 5 chick embryos. The merged image is not provided together with the single panels A-B and only appear in panel G of the same figure, this is confusing. Panel B has two scale bars. The authors tried to show *SOX2*-T co-expression by thresholding and masking *SOX2*-T signals in panel C and Suppl. Figure 1. The authors should clearly explain how this thresholding is obtained. Is this manually set? Is this set based on global or local signal intensity distribution? Otsu thresholding? In order to avoid artifacts due to thresholding, the authors should also display a scatter plot showing nuclear intensity values for *SOX2*-T. FACS analysis and quantitative assessments of the percentage of cells would further strengthen the authors' claim.

– The authors should validate *SOX2*-T expression in quail embryos

– The authors should better discuss the mechanical forces that are implicated in epithelium to mesenchyme, and in the posterior gradient of proliferation that counteracts ingression in the anterior PS epiblast. Is this affecting/related to WNT or hyppo pathway?

– Can the authors speculate about the conservation of this phenomena in the human context? What is the evidence that this conserved in human?

– The authors speculate about in-vitro generated NMPs, did they try transplantation of in-vitro generate cells? Can they compare the transcriptomic signature of the human in-vitro generated NMP with the chick counterpart?

– The tables in Suppl. Figure 4 should be a supplementary table.

– In Suppl.Fig5D there is not scale bar to assess classification efficacy and material and methods do not describe how this analysis has been performed.

– The authors should substantially revise scRNA-seq data.

*Reviewer #2 :*

In this manuscript, the authors aimed firstly to prospectively identify cells with dual neural and mesodermal fate. Previous studies in mouse had determined that such dual-fated cells existed, and had narrowed down their location to subregions of the axial progenitor zone co-expressing *Sox2* and T. However, these studies had not shown that individual cells in the *Sox2*/T positive region were dual-fated. The authors use three approaches: live imaging, and two prospective clonal labelling studies of the *Sox2*/T positive region to show, for the first time, that indeed this region harbours dual-fated cells. This is an important missing piece of the current volume of work on NMPs. Whilst the rationale for the experiments are clear, it might be better to present this work more in the context of previous studies. It seems a little harsh to dismiss population fate mapping studies as 'failing to identify' NMPs rather than complementing the clonal analysis by identifying regions potentially harbouring NMPs. In addition, there are interesting parallels with the clonal analysis carried out in mouse. For example, not all NM clones contribute to the tail bud; the observation in live imaging studies that NMPs first contribute to the neural tube and later to mesoderm is also similar to the observations from mouse retrospective clonal analysis; this could perhaps be discussed more fully.

Next, the authors carry out single cell RNA-seq analyses of three stages of chick axial elongation, and compare these with existing datasets in mouse, adding data from a later stage of mouse embryos. This aspect of the study has again confirmed and extended previous studies, showing both similarities and differences between chick and mouse NMPs. Trajectory analysis suggests progression of NMPs towards neural and mesodermal lineages in both chick and mouse. This is a valuable resource for the community. However once again it would be good to present this aspect of the work in the context of previous studies showing maturation of NMPs over time and an increased mesenchymal transcriptomic signature of NMPs at late compared to early timepoints.

Finally, the authors carry out dynamic cell tracking and show that NMPs ingress later, and proliferate faster than more posteriorly-placed cells in the primitive streak, leading to exhaustion of the posterior populations and long-term retention of the NMP population. This part of the study is well-executed and contains important new information that advances our understanding of both NMPs and axial progenitors as a whole.

Suggestions for strengthening the science

1. The results presented in Figure 1 are a little difficult to interpret as there is some background fluorescence in *Sox2* and Msgn1 staining. Perhaps the thresholding strategy could be described in the supplementary data to support statements about *Sox2* positive time of emergence and the location of Msgn1 positive/*Sox2* negative axial levels. It would be good to cite the source of the fate mapping information locating NMP/PMP/LPP/EMP in Figure 1D- is it this study or another? The section at level 5 in Figure 1E is named LPP in 1E but corresponds to a region marked 'EMP' in Fig1D. Together these two factors make it hard for the reader to judge where the posterior limit of *Sox2*/T positive cells is, relative to Msgn1 and to regions of differential fate. Also, the text (p5) says there are 'low levels [of TBXT] in the node' but the expected high levels in the node/ emerging notochord are visible in Fig1E sections 1-2.

2. In Figure 2, the clustering shows two datasets, (which are presumably batches?) of cells for each stage in chick (st5 1 etc). However, the colour coding does not allow the reader to distinguish these. Do the batches occupy different positions in the UMAP plot? How was batch correction done?

3. The cells annotated as NMPs in chick show elevated expression of Hes5 (6-somite stage) and Dll1 (combined dataset) relative to the cells annotated as PSM or neural (Figure 3, S4,5). Also, some of the NMP signature markers are expressed in a higher percentage of the NMP population in mouse compared to chick. This suggests that the cluster annotated as NMPs in chick also includes the midline primitive streak. The absence of *Sox2* from this dataset is not, in my opinion, especially troublesome as it is expressed at low levels. However, it means that the chick single cell transcriptomes cannot be independently verified as NMPs and distinguished from midline streak cells. Perhaps the mouse data can help here. Instead of using the chick as a starting point to define the NMP phenotype, if the authors can work back from cells dissected from the *Sox2*/T positive regions (ie from Dias et al., *eLife* 2020 and Gouti et al., Dev Cell 2017), as well as the NMP annotations in the mouse atlas data used here (Pijuan-Sala et al), how well do the chick cells correspond to the mouse ones? If a narrower definition of NMPs is used in chick, does this remove any of the variation seen between mouse and chick?

4. The mouse NMP trajectory analysis (Figure 3M) shows two arrows. What do they represent? if just pseudotime, it looks more like a radiation in all directions from the centre. Interestingly this would suggest NMPs also progress (upwards on the Y axis) as NMPs in pseudotime. Is this effect seen in the chick NMPs?

4. The annotations of chick versus mouse early and late clusters are confusing, In chick, there are two early clusters (coloured the same) annotated nmp-1 and nmp-2, but these do not correspond to the two earlier stages. What do they correspond to? Some discussion of the heterogeneity would be appropriate- or possibly a reanalysis of the NMP annotation in point 3 above may highlight that one of these clusters is less NMP-like. In the mouse, there is one early subcluster and two late ones. These are also annotated nmp-1 and nmp-2. A different nomenclature should be used to highlight that they are not the same as the chick ones. Unlike the chick clusters, these seem to correspond to different stages of development.

Suggestions for improving the manuscript

1. The statements that no previous studies have prospectively identified neuromesodermal-fated cells (abstract, introduction p4, discussion p10) should be modified. Forlani et al., Development 2003 10.1242/dev.00573 and Wood et al. BioRXiv https://doi.org/10.1101/622571 show dual-fated cells in mouse and chick. I think that it would be fairer in the context of previous studies to cite the population or oligoclonal studies of Iimura and Pourquié, (2006) 10.1073/pnas.0610997104, Cambray and Wilson, (2007) 10.1242/dev.02877 and Brown and Storey, (2000) 10.1016/s0960-9822(00)00601-1 as showing potential locations for cells of dual neuromesodermal fate.

2. The words 'bipotent/monopotent' used throughout the manuscript is not appropriate for the studies here, which are demonstrating fate and not potency. I would suggest 'dual fate'.

3. NMPs are presented here as 'stem cells' when the transcriptome analysis shows they mature with time. It would be good to define exactly in what sense the term is being used- it is valid since the manuscript describes an enduring progenitor producing neural and mesodermal tissue but should be used with care.

4. p5. 'We identified 7 clones containing cells expressing the same barcodes' – could this text be modified to clarify, e.g. 'expressing unique (or clone-specific) barcodes'?

5. p7 para 2 'the NMP lineage' should be changed to something not implying lineage, e.g. 'cluster'

6. Figure S7 is entitled 'NMP early and late clusters are not due to different Hox genes expression'- since the authors show differential Hox gene expression, this statement should be moderated. The text describing this observation also suggests that the genes that are differentially expressed early and late in mouse and chick are different. However, many of the genes upregulated in the later chick stage are also upregulated in mouse, eg Wnt5a, Greb1, Fgf8, Figf18, Cyp26a1 (see Wymeersch et al., 10.1242/dev.168161). It would be good to comment on this.

7. Can the authors consider whether 'persistence' of cell tracks is the optimal term? It could suggest persistence of movement rather than just their continued observation over time. Is 'longevity' a possible alternative?

---

## [Author Response]

Essential Revisions1. Many of the concerns can be addressed by text changes and minor changes to the existing data presentation as detailed in the full reviews.2. The main problems are with the bioinformatic analysis, where both reviewers highlight issues where the analysis is too superficial and needs further work.3. Additional imaging quantification is required to support their findings.

We thank the editor for the suggestions of improvement and have now thoroughly revised the text including new analyses as suggested. These are described below.

Reviewer #1:The paper entitled "Dynamics of primitive streak regression controls the fate of neuro-mesodermal progenitor cells in the chicken embryo" from Guillot and colleagues aims to characterize in the context of developing chick embryos the cellular dynamics of a population of stem cells, recently identified, called Neuro-Mesodermal Progenitors (NMPs). The authors provide several lines of evidence supporting their claims that overall look interesting, however, additional imaging quantification and more comprehensive analysis of their transcriptomic datasets are required to support their findings.Specifically, the authors generated an important scRNA-seq dataset of micro-dissect regions of the chick embryos across developmental time points from stage 5 to 35-somites. Despite the dataset represents a crucial part of the paper supporting the authors' findings of this NMP population, the analysis is not deep enough and does not report several controls and markers.

We agree with the reviewer and in the revised version, we now present a thoroughly revised analysis of the single cell RNAseq dataset as described below.

In figure 2 the authors should show heatmap of the marker genes used for classification. Louvain markers genes that define each cluster should be showed in a supplementary not as an unreadable table, instead, a dotplot or heatmap would be better. It is unclear whether Louvain clustering has been used to label the data, did the authors find that each Louvain cluster correspond to a defined cell identity?

We have now redone the entire analysis using Leiden clustering algorithm which performed better than Louvain at identifying tissue-specific clusters. As requested, we now present heatmaps showing the top 10 genes differentially expressed for each cluster in Figure 3 – Supplementary Figure 1 (for chicken) and in Figure 3 – Supplementary Figure 3 (for mouse). We have moved the gene tables originally presented in supplementary figures to a Supplementary File 2 which also includes dot plots showing the expression of genes used for identification of the clusters. We also discuss more in detail how the different clusters were identified at each developmental age in the revised text.

The pseudotime analysis is incomplete and superficial. Diffusion pseudotime strongly depends on the root cell selection. RNA velocity analysis and/or another trajectory analysis are required to better show lineage trajectory. Figure 4 L-K, the pesudotime time is not progressively ordered and it would be advisable to perform an alternative analysis to visualize lineage trajectory. Furthermore, gene expression trends should be identified and further discussed over those trajectories. The supporting information about the classification analysis using the mouse dataset is incomplete.

We now present a novel analysis of the developmental trajectories using the Waddington-OT pipeline developed by the Broad Institute (Schiebieger et al., 2019). This analysis, which is based on optimal transport, shows that the cells we identify as NMPs are indeed the most likely ancestors of both the Neural and PSM cells in the datasets. By analyzing the transport maps, this allows us to show the self-renewal of the NMP cell population and its maturation during axis elongation. We also use this tool to identify transcription factors associated with these different fates. We have added novel figures (Figure 4 and Figure 4- Supplementary Figure 1) to the revised text to describe these new results.

In addition, we have performed a Gene Set Enrichment Analysis of the early and late NMP clusters which shows an enrichment in genes associated to epithelium to mesenchyme transition in the late clusters in both species (shown in the new Figure 5). We have also completed the supporting information on the mouse dataset.

– In figure 1 the authors show SOX2-T double staining. Panels A-B-C represent the first evidence of SOX2-T co-expression in stage 5 chick embryos. The merged image is not provided together with the single panels A-B and only appear in panel G of the same figure, this is confusing.

In the revised version, we now show the merge of *SOX2* and T expression in Figure 1C instead of showing only the double-positive cells as in the previous version. We also show a blow-up of the sections of the NMP and PMP regions in Figure E’-F’ and E’’-F’’ to better illustrate the difference between these two regions which both contribute to the paraxial mesoderm as shown by the MSGN1 antibody labeling in the ingressed mesoderm.

Panel B has two scale bars.

This was corrected.

The authors tried to show SOX2-T co-expression by thresholding and masking SOX2-T signals in panel C and Suppl. Figure 1. The authors should clearly explain how this thresholding is obtained. Is this manually set? Is this set based on global or local signal intensity distribution? Otsu thresholding?

Thresholding of the *SOX2*/T cells shown in Figure 1C and Figure 1 – Supplementary Figure 1G was done manually with image J. We have now replaced the subtracted image originally shown in Figure 1C by an image showing the merge between T and *SOX2* staining which clearly identifies the double positive cells without thresholding. We have nevertheless left the subtracted image with the manual thresholding performed with image J in Figure 1 – Supplementary Figure 1G in the revised version. We also explained the thresholding in the legend of this figure and material and method section.

In order to avoid artifacts due to thresholding, the authors should also display a scatter plot showing nuclear intensity values for SOX2-T.

Since all the thresholding was performed manually, we could verify that virtually all cells in the images (shown in Figure 1- Supplementary Figure 1G) were indeed expressing both signals.

FACS analysis and quantitative assessments of the percentage of cells would further strengthen the authors' claim.

While this is an interesting suggestion, this experiment would be very challenging as the number of *SOX2*/T double positive cells is expected to be a few hundreds at the stages discussed. Thus, getting quantitative data on such a low number might be difficult. Moreover, since both markers are nuclear, this would require sorting after fixation and labeling of the cells which is much trickier than sorting for live cells labeled with a membrane marker. In addition, *SOX2* and T are expressed in overlapping opposite antero-posterior gradients in epiblast cells with significant co-expression in the epiblast adjacent to the anterior PS. The positioning of the gates for FACS analysis would be somehow arbitrary as neither marker is truly specific for the NMP population.

– The authors should validate SOX2-T expression in quail embryos

Recent work from the Benazeraf laboratory has examined the distribution of T-*SOX2* cells in the quail embryo and their distribution is similar to that we observe in the chicken embryo (Romanos et al., bioRxiv; doi: https://doi.org/10.1101/2020.11.18.388611).

– The authors should better discuss the mechanical forces that are implicated in epithelium to mesenchyme, and in the posterior gradient of proliferation that counteracts ingression in the anterior PS epiblast. Is this affecting/related to WNT or hyppo pathway?

While this is indeed an interesting question, our paper is mostly focused on the lineage and fate of NMP cells in the chicken embryo. We have no experimental data addressing the mechanical properties of ingressing cells and we are not aware of such data available in the literature for the stages considered in our study. Thus, we feel this discussion is beyond the scope of the paper.

– Can the authors speculate about the conservation of this phenomena in the human context? What is the evidence that this conserved in human?

Several groups including ours have reported that *SOX2*/T cells showing a similar identity to NMPs can be obtained from human pluripotent cells (ES/iPS) in vitro after Wnt activation. We have reanalyzed our human dataset from *SOX2*/T cells differentiated in vitro from iPS cells (day1 of Diaz-Cuadros et al., 2020) and observed that the list of human *SOX2*/T cells signature genes was much more similar to that of chicken and mouse when looking at the genes differentially expressed with the undifferentiated iPS only rather than with all the clusters as performed in the original publication. This revised list of human NMP signature genes is shown in Supplementary File 4 together with the genes conserved with chicken and mouse NMPs. We find a significant degree of conservation among the signature NMP genes in the three species and discuss these comparisons in a new paragraph in the revised text.

– The authors speculate about in-vitro generated NMPs, did they try transplantation of in-vitro generate cells? Can they compare the transcriptomic signature of the human in-vitro generated NMP with the chick counterpart?

We have not tried such transplantations of in vitro derived NMP cells and we believe this is beyond the scope of this work. Such work is however described in the following paper: “The Chick Caudolateral Epiblast Acts as a Permissive Niche for Generating Neuromesodermal Progenitor Behaviours Peter Baillie-Johnson, Octavian Voiculescu, Penny Hayward, and Benjamin Steventon”.

As suggested and discussed above, we have now compared the transcriptomic signature of cells of the chicken and mouse NMP clusters with that of the human NMP-like cells. This is now shown in Supplementary File 4.

– The tables in Suppl. Figure 4 should be a supplementary table.

We have moved the gene tables shown in supplementary figures to the Supplementary File 2 in the revised MS.

– In Suppl. Fig5D there is not scale bar to assess classification efficacy and material and methods do not describe how this analysis has been performed.

We apologize for this mistake, and we have now added it to the figure. The details of the analysis is presented in the material and method section: “Machine-learning classification of cell states”. The classification of the different cellular states was performed using an LDA classifier as described in Diaz-Cuadros et al., in vitro characterization of the human segmentation clock. Nature 580, 113-118, doi:10.1038/s41586-019-1885-9 (2020), which we now cite in the material and methods section.

– The authors should substantially revise scRNA-seq data.

As described above, we now provide a much deeper analysis of our single cell RNAseq data in the revised MS. We improved our analysis by changing our clustering method to Leiden that allows better identification of cell identities in the data. All the figures were remade with Leiden clustering.

We also provide an extended discussion of the datasets and the genes that were used to identify the various clusters in the revised text, as well as heat maps and dotplots of the genes used to identify the clusters in Figure 3 – Supplementary Figure 1 (chicken) and 6 (mouse) and in Supplementary File 2. In Figure 3F-M, we now show Diffmap plots to better illustrate the differentiation trajectories of the NMP cells.

We also performed a new analysis of the data using the Waddington-OT pipeline (Schiebieger et al., 2019). Unlike pseudotime, this analysis does not impose to select a root for the trajectories and considers the actual developmental time. This analysis shows that the NMP cells are the most probable common ancestors to the PSM and Neural cells in both chicken and mouse datasets. It also identifies a developmental trajectory in the NMP lineage which corresponds to their maturation during axis elongation. Using transport matrices, we could also quantify the contribution of NMPs to PSM, NT and NMPs and found equivalent contributions to these three lineages. This analysis therefore strongly supports the self-renewing properties of these cells. The WOT analysis also identified a number of interesting transcription factors associated with the NMP, PSM and neural fates. Expression patterns of a selection of transcription factors enriched in the NMPs is presented in Figure 4 – Supplementary Figure 1 and the extensive lists are provided in Supplementary File 3. We now also provide the expression levels of some genes involved in the NMP differentiation toward the mesodermal and neural cellular states in the new Figure 4 and in Figure 4 – Supplementary Figure 1. The results of the WOT analysis are described in a new paragraph in the revised text and shown as a new Figure 4.

We have also performed a Gene set enrichment analysis (GSEA) to compare the NMP early and late clusters identified using Leiden algorithm. This revealed an enrichment in genes associated to epithelium to mesenchyme transition in late vs early NMP clusters in both species. This analysis is now shown in Figure 5.

These new analyses led to new figures 3, 4 and 5 and Supplementary Figures 4, 5, 6, 7, 8 and 9 as well as new Supplementary Files 2, 3, 4 and 5.

Reviewer #2:In this manuscript, the authors aimed firstly to prospectively identify cells with dual neural and mesodermal fate. Previous studies in mouse had determined that such dual-fated cells existed, and had narrowed down their location to subregions of the axial progenitor zone co-expressing Sox2 and T. However, these studies had not shown that individual cells in the Sox2/T positive region were dual-fated. The authors use three approaches: live imaging, and two prospective clonal labelling studies of the Sox2/T positive region to show, for the first time, that indeed this region harbours dual-fated cells. This is an important missing piece of the current volume of work on NMPs. Whilst the rationale for the experiments are clear, it might be better to present this work more in the context of previous studies. It seems a little harsh to dismiss population fate mapping studies as 'failing to identify' NMPs rather than complementing the clonal analysis by identifying regions potentially harbouring NMPs. In addition, there are interesting parallels with the clonal analysis carried out in mouse. For example, not all NM clones contribute to the tail bud; the observation in live imaging studies that NMPs first contribute to the neural tube and later to mesoderm is also similar to the observations from mouse retrospective clonal analysis; this could perhaps be discussed more fully.

We agree and have revised the text along the lines suggested by the reviewer.

Next, the authors carry out single cell RNA-seq analyses of three stages of chick axial elongation, and compare these with existing datasets in mouse, adding data from a later stage of mouse embryos. This aspect of the study has again confirmed and extended previous studies, showing both similarities and differences between chick and mouse NMPs. Trajectory analysis suggests progression of NMPs towards neural and mesodermal lineages in both chick and mouse. This is a valuable resource for the community. However once again it would be good to present this aspect of the work in the context of previous studies showing maturation of NMPs over time and an increased mesenchymal transcriptomic signature of NMPs at late compared to early timepoints.

We agree and have now extended our analysis and compared the identified transcriptomic signature of the NMP clusters of the chicken and mouse datasets to a set of published NMP transcriptomic signatures from mouse (Gouti et al., 2017, Dias et al., 2020, Guibentif et al., 2021; Diaz-Cuadros et al., 2020) and human (in vitro derived) NMPs (Diaz-Cuadros et al., 2020). We also compare the signature of the early and late clusters identified in our analysis to early and late NMP signatures described by Gouti et al., (2017) and E8.0 NMPs from Dias et al., 2020. We also separately compare the signature of the early and late NMPs to the genes up and down regulated in small dissected regions of the NMP from the Node streak border (NSB, E8.5) and the Chordo neural hinge (CNH, E10.5) in mouse from Wymeersch et al., 2019. We added a paragraph describing these comparisons in the revised text and present these data in Supplementary Files 4 and 5.

Finally, the authors carry out dynamic cell tracking and show that NMPs ingress later, and proliferate faster than more posteriorly-placed cells in the primitive streak, leading to exhaustion of the posterior populations and long-term retention of the NMP population. This part of the study is well-executed and contains important new information that advances our understanding of both NMPs and axial progenitors as a whole.Suggestions for strengthening the science1. The results presented in Figure 1 are a little difficult to interpret as there is some background fluorescence in Sox2 and Msgn1 staining. Perhaps the thresholding strategy could be described in the supplementary data to support statements about Sox2 positive time of emergence and the location of Msgn1 positive/Sox2 negative axial levels.

We now provide higher magnification images of the T^+^/*sox2*^+^ and T^+^/*SOX2*^-^ and their corresponding MSGN1 regions as Figure 1 E’-F’ and E’’-F’’. Without thresholding, the images shown clearly illustrates the difference between the two paraxial mesoderm-producing regions which both exhibit MSGN1 positive nuclei in the mesodermal layer while *SOX2* is only seen in the epiblast of the anterior (level 3, NMP) and not in the posterior (level 4, PMP) regions. This shows that the paraxial mesoderm progenitor domain can be subdivided in 2 progenitor domains both producing paraxial mesodermal cells (MSGN1+).

It would be good to cite the source of the fate mapping information locating NMP/PMP/LPP/EMP in Figure 1D- is it this study or another?

The fate mapping used was from Psychoyos and Stern, (Development, 1996) who generated a very detailed fate map of the PS of the chicken embryo encompassing the stages described here. We have added this information to the revised text.

The section at level 5 in Figure 1E is named LPP in 1E but corresponds to a region marked 'EMP' in Fig1D. Together these two factors make it hard for the reader to judge where the posterior limit of Sox2/T positive cells is, relative to Msgn1 and to regions of differential fate.

We apologize for the oversight. This study is focused on the fate of paraxial mesoderm and we made no attempt to distinguish between the lateral plate and extraembryonic mesoderm territories of the posterior primitive streak. We have now relabeled the posterior primitive streak territory as LPP/EMP on Figure 1E. This corresponds to the territory devoid of MSGN1 expression.

Also, as a minor point, the text (p5) says there are 'low levels [of TBXT] in the node' but the expected high levels in the node/ emerging notochord are visible in Fig1E sections 1-2.

It is indeed right that the node shows high TBXT expression but the level of TBXT in the epiblast around the Node is low as seen in our quantifications and transverse sections in Fig1E sections 1-2. We meant 'low levels [of TBXT] in epiblast around the node’ and have changed the text accordingly.

2. In Figure 2, the clustering shows two datasets, (which are presumably batches?) of cells for each stage in chick (st5 1 etc). However, the colour coding does not allow the reader to distinguish these. Do the batches occupy different positions in the UMAP plot? How was batch correction done?

The 2 datasets are technical replicates. We agree that the color coding is not properly showing the heterogeneity of the data and we now provide a Figure 3F and in Figure 3 – Supplementary Figure 1 with a new set of colors. As seen in the new figures, the 2 datasets do not occupy different parts of the UMAP. In most cases, we used the module Bbknn for batch correction, but for the WOT analysis, we used the Combat module. We have added this information to the M&Ms section.

3. The cells annotated as NMPs in chick show elevated expression of Hes5 (6-somite stage) and Dll1 (combined dataset) relative to the cells annotated as PSM or neural (Figure 3, S4,5). Also, some of the NMP signature markers are expressed in a higher percentage of the NMP population in mouse compared to chick. This suggests that the cluster annotated as NMPs in chick also includes the midline primitive streak. The absence of Sox2 from this dataset is not, in my opinion, especially troublesome as it is expressed at low levels. However, it means that the chick single cell transcriptomes cannot be independently verified as NMPs and distinguished from midline streak cells.

This is correct, as in our immunostaining we see that T-*SOX2* double positive cells are also present in the anterior primitive streak (in the groove) at stage 5HH (see Figure 1C and 1E level 3). This suggests that the distribution of NMPs might be slightly different in chicken compared to mouse embryos. This is supported by the observation that we cannot identify any PS specific cluster, although we cannot rule out that this due to the limited resolution of the dataset.

Perhaps the mouse data can help here. Instead of using the chick as a starting point to define the NMP phenotype, if the authors can work back from cells dissected from the Sox2/T positive regions (ie from Dias et al., eLife 2020 and Gouti et al., Dev Cell 2017), as well as the NMP annotations in the mouse atlas data used here (Pijuan-Sala et al.,), how well do the chick cells correspond to the mouse ones? If a narrower definition of NMPs is used in chick, does this remove any of the variation seen between mouse and chick?

We appreciate the suggestion which in principle could be a good way to present the data. The problem we are facing here, which we did not sufficiently highlight in the previous version of the paper, is that chicken cells were sequenced with Indrops, which provides limited depth, whereas mouse cells were sequenced with SMARTseq (Gouti et al.,) or 10x (Pijuan-Sala, Dias et al.,) which allows for more sequencing depth. This is probably the main reason why we do not identify *Sox2* or Nkx1.2 in our chicken dataset. This is also why it is difficult to identify the chicken NMPs based on the mouse NMP signature genes. However, the most stringent definition of NMPs is not based on expression of marker genes (in fact, we don’t even know whether *SOX2*/T cells are NMPs) but is based on their ability to give rise to both neural tube and PSM. This is the criterion we use to define NMPs in this analysis. Our new WOT analysis strongly argues that in both chicken and mouse, cells of the identified NMP clusters are ancestors to the PSM and Neural Tube.

We now present a comparison between the NMP signature we identified in chicken and mouse embryos and lists of genes identified as NMP signatures in mouse embryos and human NMP-like cells differentiated in vitro in the following papers: Gouti et al., Dev Cell 2017; Wymeersch et al., 2019; Dias et al., *eLife* 2020; Guibentif et al., Dev Cell, 2021, Diaz-Cuadros et al., Nature 2020. This analysis shows significant conservation of genes between these different lists. While only 38 genes are conserved in 4 or 5 of the 6 datasets compared. The most striking characteristic of this list is the strong enrichment in genes of the Wnt pathway, consistent with the requirement for Wnt signaling for NMP differentiation in vivo and in vitro. We also find an enrichment of genes involved in glycolysis in line with recent papers describing higher glycolytic activity required for Wnt signaling in the tail bud (Oginuma et al., 2017, 2020; Bulusu et al., 2017). This new analysis is detailed in a new paragraph in the revised text and described in Supplementary File 4 and 5.

4. The mouse NMP trajectory analysis (Figure 3M) shows two arrows. What do they represent? if just pseudotime, it looks more like a radiation in all directions from the centre. Interestingly this would suggest NMPs also progress (upwards on the Y axis) as NMPs in pseudotime. Is this effect seen in the chick NMPs?

As suggested, we now present diffusion maps plots in Figure 3 that better show the pseudotime trajectories toward the neural and mesodermal fates in chicken and mouse cells. Indeed, NMP also exhibit changes in pseudotime that parallel developmental time in both species although this is more subtle in chick than in mouse probably because of sequencing depth. We also performed a new analysis of the mouse and chicken datasets using the Waddington OT pipeline (Schiebieger et al., 2019). This analysis shows that the NMPs are the most probable ancestors of the PSM and neural cells. It also identifies a trajectory in the NMP cluster (confirming the trajectories observed in pseudotime) correlating with the age of the samples, suggesting maturation of these cells. We now describe these new analyses in detail and have added a new figure 4 to present this data.

4. The annotations of chick versus mouse early and late clusters are confusing, In chick, there are two early clusters (coloured the same) annotated nmp-1 and nmp-2, but these do not correspond to the two earlier stages. What do they correspond to? Some discussion of the heterogeneity would be appropriate- or possibly a reanalysis of the NMP annotation in point 3 above may highlight that one of these clusters is less NMP-like. In the mouse, there is one early subcluster and two late ones. These are also annotated nmp-1 and nmp-2. A different nomenclature should be used to highlight that they are not the same as the chick ones. Unlike the chick clusters, these seem to correspond to different stages of development.

The analysis of the early and late NMP clusters has been revisited using Leiden clustering (instead of Louvain initially). We now only find one early and late clusters in both species. We perform a GSEA analysis using the gene set “Epithelium to mesenchyme transition” which shows enrichment in the late clusters in both species. We also compared these early and late clusters to other similar datasets published in the literature (Gouti et al., 2017; Wymeersch et al., 2019; Dias et al., 2020; Dias-Cuadros et al., 2020). This part has been modified in the revised text and a new figure 5, Supplementary File 4 and Figure 5 – supplementary Figure 1 have been generated with the new data.

Suggestions for improving the manuscript1. The statements that no previous studies have prospectively identified neuromesodermal-fated cells (abstract, introduction p4, discussion p10) should be modified. Forlani et al., Development 2003 10.1242/dev.00573 and Wood et al., BioRXiv https://doi.org/10.1101/622571 show dual-fated cells in mouse and chick.

We are well aware of the Forlani paper, where they perform single cell injections of HRP in epiblast cells. However, this study was intended for a different purpose (comparing the anterior boundaries of the clones with the timing of Hox gene activation). It is true that they show a few bipotent clones in their Figure 7 but this work was done before the identification of the NMPs in 2009 and in no place in the text do they refer to bi-potential precursors. We nevertheless introduced and discussed this reference in the text.

With respect to the Wood preprint, I don’t think they demonstrate the existence of dual-fated cells. Their data arguing for the dual fate is shown in Figure 5a-c which shows cell fate tracked for only 15 hours (ie up to the 5-somite stage approximately). Their designation of the neuro-mesodermal region is quite arbitrary. The main argument in support of a dual fate of these cells is that their descendants span neural and mesodermal territories after 15 hours when backtracked from time lapse movies (white bars on Figure 5c). However, at these stages, these two territories are not clearly separated into neural tube and paraxial mesoderm and thus the designation of the final fate is not supported by any anatomical or molecular landmark. This study is extremely preliminary and lacking critical details to evaluate the findings. I don’t think it would be fair to consider that this work (which is essentially contemporary to ours) first identified dual-fated cells in the chicken embryo. We nevertheless discuss this reference in the revised version of the manuscript.

I think that it would be fairer in the context of previous studies to cite the population or oligoclonal studies of Iimura and Pourquié, (2006) 10.1073/pnas.0610997104, Cambray and Wilson, (2007) 10.1242/dev.02877 and Brown and Storey, (2000) 10.1016/s0960-9822(00)00601-1 as showing potential locations for cells of dual neuromesodermal fate.

We were already discussing these studies in the previous version of this paper but have rewritten this part. We were not trying to diminish the merit of these oligoclonal studies, which are very elegant and compelling. Nevertheless, we believe it is fair to say that due to their design, these studies cannot prospectively identify bipotential cells, only territories.

2. The words 'bipotent/monopotent' used throughout the manuscript is not appropriate for the studies here, which are demonstrating fate and not potency. I would suggest 'dual fate'.

We respectfully disagree about using dual fated to describe NMP cells as dual fated means that the two fates can be observed. Thus, we believe using this term would only be appropriate for the description of a subset of the cells identified by the lineage analysis (and we do use it in some places) as in these experiments we can visualize the fate of the cells. However, in many cases, only one of the fates is realized even though the precursors (the NMPs) appear to be bipotent at the population level. For the scRNAseq analysis, the cells of the NMP cluster cannot be called dual-fated as the analysis does not reveal the actual fate of the cells. We believe the cells identified as NMP using this approach should be called bipotent.

3. NMPs are presented here as 'stem cells' when the transcriptome analysis shows they mature with time. It would be good to define exactly in what sense the term is being used- it is valid since the manuscript describes an enduring progenitor producing neural and mesodermal tissue but should be used with care.

We define NMPs as stem cells because they can give rise to differentiating progeny and self-renew. We now add further evidence for a self-renewing behavior with the WOT analysis transport maps which show that NMP contribute significantly to the NMP population itself over time. This is now shown in the revised Figure 4C-D. The fact that NMPs can also mature is a situation frequently observed for other stem cell types (fetal vs adult satellite cells or hematopoietic cells for instance). We have however removed stem cell from the title of the revised paper.

4. p5. 'We identified 7 clones containing cells expressing the same barcodes' – could this text be modified to clarify, e.g. 'expressing unique (or clone-specific) barcodes'?

This has been modified accordingly.

5. p7 para 2 'the NMP lineage' should be changed to something not implying lineage, e.g. 'cluster'

We agree and have removed NMP lineage from the revised text and replaced it by the appropriate clusters.

6. Figure S7 is entitled 'NMP early and late clusters are not due to different Hox genes expression'- since the authors show differential Hox gene expression, this statement should be moderated. The text describing this observation also suggests that the genes that are differentially expressed early and late in mouse and chick are different. However, many of the genes upregulated in the later chick stage are also upregulated in mouse, eg Wnt5a, Greb1, Fgf8, Figf18, Cyp26a1 (see Wymeersch et al., 10.1242/dev.168161). It would be good to comment on this.

We agree and have changed the title of what is now Figure 5 – supplementary Figure 1 with “Characterization of the early and late NMP clusters”. We have also performed a series of comparisons between the differentially expressed genes from the early and late NMP clusters identified in our chicken and mouse datasets and with the datasets from Gouti et al., 2017, Wymeersch et al., 2019, Dias et al., 2020 and Dias-Cuadros et al., 2020. We do find similarities between the gene signatures identified in our mouse and chicken early and late NMPs clusters and those identified in these studies and the results are now shown in the Supplementary File 4.

7. Can the authors consider whether 'persistence' of cell tracks is the optimal term? It could suggest persistence of movement rather than just their continued observation over time. Is 'longevity' a possible alternative?

We have made the corresponding change.